# Colchicine-Binding Site Inhibitors from Chemistry to Clinic: A Review

**DOI:** 10.3390/ph13010008

**Published:** 2020-01-03

**Authors:** Eavan C. McLoughlin, Niamh M. O’Boyle

**Affiliations:** School of Pharmacy and Pharmaceutical Sciences, Trinity Biomedical Sciences Institute, Trinity College Dublin, D02 Dublin, Ireland; niamh.oboyle@tcd.ie

**Keywords:** anti-cancer, anti-tubulin, tubulin binding, microtubule-targeting agents, colchicine, colchicine binding site, combretastatin, CA-4, tubulin destabilizing

## Abstract

It is over 50 years since the discovery of microtubules, and they have become one of the most important drug targets for anti-cancer therapies. Microtubules are predominantly composed of the protein tubulin, which contains a number of different binding sites for small-molecule drugs. There is continued interest in drug development for compounds targeting the colchicine-binding site of tubulin, termed colchicine-binding site inhibitors (CBSIs). This review highlights CBSIs discovered through diverse sources: from natural compounds, rational design, serendipitously and via high-throughput screening. We provide an update on CBSIs reported in the past three years and discuss the clinical status of CBSIs. It is likely that efforts will continue to develop CBSIs for a diverse set of cancers, and this review provides a timely update on recent developments.

## 1. Microtubules as a Target for Treating Cancer

Microtubules are major components of the eukaryotic cytoskeleton conserved across evolution. They are involved in cellular motility, intracellular transport, cell structure and maintenance of cell structure. Most importantly, in the context of anti-cancer small-molecule drug development, microtubules are responsible for the separation of chromosomes during mitosis [1]. Mitosis is the fifth phase of a typical cell cycle in which DNA is divided into two daughter cells. Mitosis is an important target for anti-cancer agents due to loss of normal cell cycle controls in malignant cells. Cancer cell hallmarks include alterations which result in unscheduled and uncontrolled proliferation in addition to genomic instability. Numerous anti-cancer strategies have focused on targeting the rapid multiplication of cancer cells in order to arrest the cell cycle and specifically kill cancer cells [2].

Microtubules are described as ‘dynamic polymers’ and are composed of tubulin heterodimers formed from α and β tubulin monomers. Tubulin is a globular protein that exists as an αβ heterodimer and is the principle building block of microtubule structures. α and β Tubulin are both 50 kDa in size and share 40% identity in amino acid homology.

Each microtubule consists of α and β tubulin heterodimers assembled together, associating in head-to-tail format to make linear protofilaments. Thirteen protofilaments associate laterally and wind together to form a 24 nm diameter, long, hollow cylinder known as the microtubule: a polar structure with a “+ end” (fast growing) and a “− end” (slow growing) (Figure 1) [3].

Microtubules alternate between periods of growth and shrinkage in what is known as ‘dynamic instability’. This occurs through the addition of guanosine triphosphate (GTP)-bound tubulin at the positive pole and the removal of guanosine diphosphate (GDP)-bound tubulin at the negative pole. Tubulin heterodimers bind GTP reversibly at the β-subunit. GTP is hydrolysed to GDP during polymerisation, which leads to loss of a GTP cap. Microtubules also undergo periods of polymerisation and depolymerisation, continually assembling and disassembling inside the cell. The phase of polymerisation versus depolymerisation is determined by the rate of the addition of GTP-bound tubulin. β-Tubulin binds GTP and in doing so regulates the rate of polymerisation of heterotubulin heterodimers. GTP hydrolysis to GDP weakens tubulin’s binding affinity for adjacent tubulin molecules and favours depolymerisation of microtubule structures. As long as GTP-bound tubulin units are added more rapidly than GTP is hydrolysed to GDP, a GTP cap at the positive pole is retained and microtubules will continue to grow. If the rate of GTP-bound tubulin slows, GTP at the “+ end” is hydrolysed, resulting in microtubule catastrophe and rapid depolymerisation [4]. 

The dynamics of microtubules in chemical equilibrium between an intracellular pool of αβ-heterodimers and large microtubule polymers are tightly regulated and central to their biological functions. In dividing cells undergoing mitosis, the function of the microtubule is to form a mitotic spindle. When cells enter mitosis, the rate of microtubule growing and shortening increases up to 100-fold. This makes such cellular structures attractive targets for chemotherapy and anti-cancer drug development. These rapid dynamics are highly sensitive towards modulation by antimitotic agents [5]. The majority of microtubule-binding agents (MTAs) bind to β-tubulin in the αβ heterodimer and suppress microtubule dynamics. This causes a delay or blockade at the metaphase-anaphase transition during mitosis [5]. Due to the disruption at the mitotic spindle, spindle cell assembly check points are disrupted, causing extended mitotic arrest and cell death [6]. 

Microtubules are the targets of chemically diverse MTAs which have been used with great success in the treatment of cancer. Drugs that disrupt microtubule dynamics are widely used in cancer chemotherapy. The vast majority act by binding to tubulin. MTAs both perturb mitosis and arrest cells during G_2_/M phase of the cell cycle. MTAs interact with tubulin at a number of different binding sites: the laulimalide, maytansine, taxane/epothilone, vinca alkaloid and colchicine sites (Figure 2) [7]. Such agents are broadly categorised into two groups when used in high concentrations: microtubule-stabilizing agents including the taxanes, epothilones, and laulimalide and microtubule-destabilising agents inclusive of colchicine, the vinca alkaloids, and maytansine. This review will focus solely on microtubule-destabilising agents binding to the colchicine-binding site (CBS) of tubulin, termed colchicine-binding site inhibitors (CBSIs).

## 2. Colchicine: Historical Uses, Structure and Mechanism of Action

Colchicine (**1**, Figure 3) is a tricyclic alkaloid (alkaloid nitrogen highlighted in blue, Figure 3) that is extracted from the plant *Colchicum autumnale*, also known as the autumn crocus or ‘poisonous meadow saffron’. Colchicine has been used as a medicine throughout history, with the first therapeutic use of *Colchicum autumnale* described almost 3000 years ago. It is a highly toxic anti-cancer MTA and effectively inhibits cellular mitosis. Colchicine may also modify voltage-dependent anion channels of mitochondrial membranes, thereby limiting mitochondrial metabolism in cancer cells [8]. Therapeutic efficacy has been established for colchicine in the treatment of acute flares of gout and gouty arthritis [9,10]. It is also used to treat familial Mediterranean fever [11], Bechet’s disease [12] and recurring pericarditis with effusion. Experimental medical use includes treatment of acute coronary syndromes [13]. This utilises colchicine’s inhibition of a nucleotide-binding domain (NOD)-like receptor protein 3 (NLRP3) inflammasome protein complex, a novel pathway independent of colchicine’s effects as an MTA. This suggests that colchicine has additional effects in addition to its potent antimitotic potential [14,15]. Although a cheap chemotherapeutic option, colchicine’s clinical applications are limited by its toxicity, narrow therapeutic index and the development of multi-drug resistance (MDR). Toxicity includes neutropenia, gastrointestinal upset, bone marrow damage and anaemia. Colchicine’s toxicity is due to its antimitotic properties. It binds to tubulin in non-cancerous cells and causes impaired protein assembly, mitotic arrest and multi-organ dysfunction [16].

Monomers of tubulin are divided into three functional domains: an amino terminal domain containing the GTP/GDP nucleotide binding region, an intermediate domain and a carboxy terminal domain [17]. Colchicine induces assembly independent GTPase activity to promote loss of the microtubule GTP cap and disassembly [18]. Upon colchicine binding to tubulin, the straight conformation of the αβ-tubulin heterodimeric subunits is lost, resulting in curved tubulin heterodimers. Lateral contacts of adjacent αβ-subunits necessary to maintain interactions between them are lost and, as lateral contacts decrease, microtubules disassemble. A steric clash between colchicine and α-tubulin inhibits microtubule assembly [19]. 

The CBS lies in the intermediate domain at the interface of α and β subunits (Figure 2). It was first characterised in 2004 by Ravelli et al. using X-ray crystallography. A 3.5 Å X-ray structure of αβ-tubulin in complex with N-deacetyl-N-(2-mercaptoacetyl) colchicine (DAMA-colchicine) clearly outlined the structure of the binding pocket which has a width of 4–5 Å. The volume of the CBS site is mediated by helix 7 (H7) containing Cysβ241, loop 7 (T7) and helix 8 (H8). Experimental data demonstrates that colchicine binds to β-tubulin at its interface with α-tubulin, subsequently inhibiting tubulin polymerisation. The important trimethoxyphenyl group of colchicine is orientated within β-tubulin close to Cysβ241 [19].

Colchicine binds with high affinity to the β-subunit. It is surrounded by mainly β-tubulin through helix 7. Cys-241 hydrogen bonds with the trimethoxyphenyl ring of colchicine while Thr-179 and Val-181 within α-tubulin form hydrogen bonds with the tropolone ring [20]. There are stringent structural requirements for binding of colchicine to tubulin which have been extensively studied. Structurally the molecule consists of a 3,4,5-trimethoxyphenyl ring (the A ring), a saturated seven membered ring containing an acetamido group at position 7 (the B ring), and a tropolone ring (the C ring, **1**, Figure 3). It has been shown that the strength of colchicine binding to tubulin is provided by the interaction of ring A with the CBS, whereas inhibition is modulated by interactions between the oxygen atoms on ring C and the CBS. It is proposed that ring A anchors and maintains the B and C rings in correct orientation within the binding locus. When the A ring structure is modified and anchoring is weakened, the free energy used to stabilise the complex in proper orientation may weaken its inhibitory effect on tubulin [18]. Changing the structure of colchicine’s, a ring results in loss of binding affinity. B and C ring structural modifications are however possible [21].

The keto functional group on Ring C and the methoxy functional groups in Ring A and Ring C are crucial for binding of colchicine (highlighted in red, **1**, Figure 3). Swapping the C ring keto and methoxy leads to inactive compounds. Ring C may be replaced with a regular benzene ring without loss of activity. Ring B’s acetamido group is not essential for activity although the stereochemistry at this carbon is important likely due to the effect of this substituent on holding the molecule in overall conformation with respect to rings A and C [22].

## 3. Targeting the Colchicine-Binding Site for Anti-Cancer Therapy

Colchicine has been widely used as a research tool in the study of microtubules. Colchicine is known to stabilise microtubule structures at low concentrations and destabilise them at high concentrations bearing a similar mechanism of action to anti-cancer drugs, e.g., the vinca alkaloids. Drugs targeting the CBS have recently appeared on the market and it is anticipated that others will follow. There are several advantages of targeting the CBS, including the inhibition of angiogenesis and overcoming multidrug resistance (MDR).

Angiogenesis is one of the hallmarks of cancer. Angiogenesis plays a critical role in cancer as solid tumours require a blood supply with adequate oxygen and nutrients to grow beyond a few millimetres in size [23]. In recent years, emerging information on the environment of solid tumours has led to the consensus that there are several advantages of developing cancer therapies to target cancer cells indirectly by attacking the tumour vasculature [24,25]. Such compounds are called vascular disrupting agents (VDAs). VDAs target the tumour’s established vasculature leading to haemorrhagic necrosis and subsequently cell death [26]. Microtubules are important regulators of endothelial cells and vasculature network formation and hence the CBS is an attractive target for VDAs. Colchicine’s anti-vascular effects are often only observed at doses approaching or exceeding the maximum tolerated dose. Given the widespread clinical use of the taxanes and vinca alkaloids, research efforts are continuously focusing on developing colchicine-like VDAs as cancer treatments. 

MDR is partially mediated by the overexpression of the ATP-dependent drug efflux pump, P-glycoprotein (P-gp). P-gp transports drugs out of cancer cells and away from their intended target. MDR is a major impediment to successful chemotherapy of human tumours. One of the major limitations of clinically used MTAs is the overexpression of the *MDR1* gene encoding for P-gp in many tumour cells; it decreases intracellular drug levels and limits efficacy of cytotoxic agents. The overexpression of P-gp is correlated to poor response and treatment failure while using the taxanes and vinca alkaloids [27]. Additionally, the expression of a mutant β-tubulin III isoform is an indicator of resistance to paclitaxel and vinorelbine. CBSIs may remain unaffected by the expression pattern of β-tubulin [28]. The use of colchicine as an anti-cancer derivative is currently excluded based on its extreme toxicity, but this property does not exclude use of its scaffold as a template for generation of potent anti-cancer derivatives. Agents targeting the CBS are attractive as they are often effective in drug resistant models, as outlined throughout this review. They would provide another treatment option for cancer patients at a fraction of the cost of biologicals such as monoclonal antibodies.

## 4. Colchicine-Binding Site Inhibitors Known before 2017

### 4.1. Combretastatin A-4 and Analogues

The combretastatins are a group of diaryl stilbenoid (diaryl stilbene highlighted in blue, **2**, Figure 3) natural products which were isolated by Pettit et al. in 1989 from the bark of the South African bush willow tree *Combretum caffrum* [29]. *Combretum caffrum* belongs to Combretaceae family of shrubs found in South Africa, mainly of the Eastern Cape and Transkei to Natal. It was used for decades as a remedy for ailments ranging from heart and worm remedies, wound dressings and scorpion stings by the Xhosa tribe in South Africa [30]. Pettit et al. demonstrated that isolates from *Combretum caffrum* bark were active against murine lymphocytic leukaemia cell lines (P-388). This antiproliferative activity was later attributed to the combretastatin A series of compounds. In total, among 17 original natural combretastatins isolated, two *cis* stilbene compounds have attracted profound interest over the years. Combretastatin A-4 (CA-4, **2**, Figure 3) and combretastatin A-1 (CA-1, **3**, Figure 3) are the most potent analogues and have been the focus of much attention in areas of chemistry, biochemistry and clinical application. CA-1 may have enhanced potency relative to CA-4 due to its anti-vascular effects but also due to its potential to induce cancer cell death via an *ortho*-quinone mechanism, binding to cellular nucleophiles and forming free radicles [31].

The *cis* stilbene CA-4 has proven itself to be a significant cancer cell growth inhibitor and antimitotic anti-cancer agent through the inhibition of tubulin polymerisation via binding at the CBS [32,33]. CA-4 is structurally related to colchicine, containing two phenyl rings tilted at 50–60° to each other, linked in *cis* geometry by a double bond and containing a 3,4,5-trimethoxyphenyl (highlighted in blue of CA-1, **3**, Figure 3) ring system. It binds in similar orientation as colchicine at the CBS. Both CA-4 and CA-1 and their respective phosphate prodrug salts, fosbretabulin (**83**, see the figure in Section 6.1.1) and Oxi4503 (**84**, see the figure in Section 6.1.1) are of increasing therapeutic interest. CA-4 shows promise in comparison to CA-1 in the potential treatment of tumours with acquired drug resistance [34].

CA-4 is not only considered to be an antimitotic agent. In addition to its antiproliferative effects as an inhibitor of tubulin polymerisation, it also has potent anti-vascular and anti-angiogenic effects categorising it as a VDA. In 1997, Watts et al. noted an increase in endothelial permeability with CA-1. This indicated that the compound was targeting tumour vasculature and leading to cell death from ischaemic and tumour haemorrhagic necrosis. This selective disruption of tumour associated microvessels results in loss of nutrients, oxygen deprivation and ultimately necrosis [33]. The ability of typical MTAs namely vinblastine and colchicine to inhibit tumour growth via vascular mechanisms is well established [35]. Colchicine is not currently in clinical use as an anti-cancer treatment due to colchicine’s narrow therapeutic index [36]. The phosphate derivative of CA-4P was found to inhibit tumour blood flow at concentrations 10-fold less than its maximum tolerated dose, which led to the first clinical trial of CA-4 as a VDA [37].

CA-4 appears to disrupt endothelial cell-specific junctional molecule vascular endothelial cadherin (VE-cadherin), which inhibits endothelial cell migration and capillary tube formation. The morphological changes induced on the endothelial cells lining micro vessels proceed to destroy the tumour from within [38]. CA-4 causes occlusion of the tumour vasculature, resulting in hypoxia-induced necrosis, which targets the core of solid tumours [39]. This mechanism is unlike many other anti-vascular treatments which generally target peripheral but not core tumour cells, resulting in MDR. CA-4 appears to be specific for tumour endothelial cells, although the precise mechanism for selectivity over normal endothelial cells remains unclear. VDAs for cancer treatment are an attractive option as angiogenesis occurs in only a limited number of situations, for example, wound healing and during the menstrual cycle. Therefore, anti-angiogenic strategies hold potential for specificity with little toxicity. 

CA-4’s potential as an anti-cancer agent has long since been recognised, and it demonstrates potent cytotoxicity against MDR cancer cell lines [38,40]. Resistance to CA-4P monotherapy is seen however in the tumour rim [41]. This is likely due to additional support structures seen in mature vessels provided by the actin cytoskeleton, mature basement membranes and vessel-associated pericytes. This protects them from tubulin-binding disruption by VDAs. This suggests that combretastatin derivatives are ideal candidates to complement existing anti-cancer approaches, given that cells of the mature vasculature are more susceptible to conventional chemotherapy strategies [38]. Indeed, successes have been made in the clinical setting by combining combretastatin prodrugs with established therapy as will be discussed below in the section of this review on clinically used CBSIs. 

### 4.2. Successful Modifications of Combretastatin A-4

CA-4 is a very potent anti-cancer agent. Its structural simplicity renders it amenable to chemical manipulation with the aim of modifying solubility, stability and therapeutic efficacy. CA-4 is typically observed to inhibit 50% of cancer proliferation within a dose range of 1–10 nM. It inhibits microtubule polymerisation in a dose range of 2–3 μM. Many promising CA-4 analogues have analogous cytotoxicity and microtubule inhibitory properties. Thousands of analogues have been synthesised and described. CA-4 (2) is a fascinating lead molecule in design of CBSIs. This review aims to focus attention on recently emerging research on CA-4 derivatives with strong potential for further development as clinically used anti-cancer agents. 

CA-4 can exist in two geometric configurations—*cis* and *trans* stilbene. The *cis* structure has been shown to be important for binding to the CBS. Only the *cis* configuration of CA-4 and other stilbene derivatives possess anti-cancer bioactivity. The active *cis* conformation of CA-4 readily isomerizes into the more thermodynamically stable but significantly less active *trans* isomer, demonstrating little or no biological activity [42]. Isomerization of *cis* CA-4 is readily observed on storage and additionally in vivo during metabolism accompanied by a dramatic reduction in both anti-tubulin and anti-tumour activity [43,44] recently reported due to structural distortion of *trans* CA-4 in the CBS [45]. 

CA-4 consists of three important structural features as briefly mentioned above. SAR studies have demonstrated that the following are crucial for tubulin polymerisation inhibitory effects: (1)3,4,5-trimethoxyphenyl-subsituted ring A.(2)3-hydroxy-4-methoxyphenyl-substituted ring B.(3)*cis* double bond separating the two phenyl rings.

It is well known that the insertion of a heteroatom as a bioisostere in place of a carbon atom can yield new systems with modified pharmacodynamic and pharmacokinetic properties, including lipophilicity, polarity, aqueous solubility and importantly potency and selectivity [46]. A number of heterocyclic bridging CA-4 analogues have been prepared to restrict the *cis* configuration and provide optimal conformational geometry for interaction with the CBS. Oshumi et al. were the one of the first to replace the olfenic bond with a series of heteroatoms, prompted by observations upon superimposition of CA-4 and colchicine structures. A good match was seen, suggesting that the double bond region may in fact be amenable to expansion. A compound with a 1,2,4-triazole replacing the *cis* double bond (**4**, Figure 3) retained both cytotoxic and anti-tubulin activities, a significant advancement as the first successful replacement of the double bond by a ring structure with cytotoxic activity. 

Synthesised combretastatin heterocycle derivatives to date include thiazoles, imidazoles, pyrroles, oxazoles, and many others [47]. Due to the increase in hydrophilicity of compounds containing lone pair of electrons, upon the introduction of such heterocyclic moieties, these molecules have demonstrated improved water solubility with respect to CA-4 [48]. Many CA-4 analogues are highly potent tubulin inhibitors with nanomolar or picomolar antiproliferative IC_50_ values. However, certain *cis*-restrained analogues have reduced activity. Analysis of their structures can provide useful insight into why this may be the case. Three cases of structures that do not appear to be optimal for binding to the colchicine site of tubulin are:(1)Three-membered ring bridges,(2)Six-membered ring bridges, and(3)Immobilized trimethoxyphenyl moieties.

In an example of the first case, Fürst et al. replaced CA-4’s double bond with a series of cyclopropyl units in an attempt to maintain the desirable *cis*-diaryl structural relationship. However, these compounds were significantly less active than corresponding stilbene derivatives. The cyclopropyl of CA-4 was found to have an IC_50_ of 0.028 μM and 0.102 μM in Hela and MCF-7 cells respectively compared to 0.00051 μM and 0.0025 μM of CA-4. Despite low in vitro activity as an anti-cancer agent, cyclopropanes held less tendency to undergo *cis* to *trans* isomerization [49]. This illustrates that three membered ring bridges do not appear to be optimal for binding to the CBS. Secondly, replacing the double bond with a six-membered ring in all cases resulted in a large decrease in cytotoxicity. A good illustration is the replacement of an imidazole (**5**, Figure 3, highlighted in blue) ring with a pyrazine (**6**, Figure 3, highlighted in blue) ring resulted in a loss of 350 fold potency in non-small-cell lung carcinoma cell lines, with a dramatic increase in IC_50_ in NCI-H640 cells from 2 μM (5) to 707 μM (6) [50]. Thirdly, rigidifying the combretastatin trimethoxyphenyl moiety was an interesting possibility explored. Paracyclophane derivatives (e.g., **7**, Figure 3) were synthesised, restricting the conformational freedom of the aryl rings with the polyether linkage with a polyether linkage between the A and B ring. These compounds did not possess any anti-angiogenic or biological activity [51]. These modifications provide important information on the structural tolerance of the CBS. 

This review aims to highlight the most recent and exciting data for compounds with superior therapeutic efficacy to CA-4, discussing literature reports of structural modifications to CA-4 from 2017 to 2019 and selected others. Modifications of CA-4 have produced many synthetic derivatives with extremely potent antiproliferative and anti-tumour effects. A number of other reviews of CA-4 analogues have been published discussing prior modifications [3,52,53,54,55].

#### 4.2.1. Potent Chalcone Derivatives of CA-4

Ducki et al. [56,57] used a chalcone scaffold to synthesise keto stilbene derivatives of CA-4 (**8**–**12**, Figure 4). They incorporated the aryl substitution pattern of CA-4 into their chalcones to obtain several compounds with potent in vitro antiproliferative activity at nanomolar concentrations against human chronic myelogenous leukaemia K562 cell lines, superseding the activity of CA-4. The introduction of an α-alkyl substituent to the enone improved potencies up to 20-fold. Compound **8** was roughly 200-fold less potent than **9**, inhibiting cancer cell proliferation in K562 at an IC_50_ of 4.3 nM versus 0.21 nM for **9**. The introduction of an alkoxy, e.g., SD400 (**10**, Figure 4), also increased the potency (IC_50_ = 1.5 nM). The α-methoxy chalcone 10 was identified as a potent inhibitor of tubulin assembly with an IC_50_ of 0.21 nM superior to CA-4’s IC_50_ of 2.0 nM. K562 cells treated with these chalcones are blocked in mitosis consistent with classical CBSIs. The chalcones displaced colchicine from its binding site, confirming them as CBSIs. Compounds **8**–**12** have better activity in vivo than CA-4 in terms of vascular damaging properties. Conformational preferences for enone systems are generally s-*trans* unless forced into an s-*cis* arrangement by steric hindrance. Chalcone **8** adopts an s-*cis* conformation whereas the α-methyl chalcone **9** adopts the s-*trans*, suggesting that s-*trans* is good for bioactivity. The two aryl rings of chalcone **9** superimpose onto those of CA-4 better than that of **8** and therefore it is hypothesised that the introduction of the α-methyl results in stronger binding in the CBS. Despite reports of pre-clinical studies for the phosphate prodrugs **11** and **12** (Figure 4), these compounds have not advanced into clinical studies [56]. 

#### 4.2.2. Phenstatin and Derivatives 

Diarylketone phenstatin (**13**, Figure 4) is a CA-4 analogue discovered serendipitously, where CA-4’s olefin is replaced with a carbonyl group (highlighted in blue, Figure 4). It was obtained unexpectedly following Jacobsen oxidation; during an attempted epoxidation of CA-4 silyl ether to give the silyl ether of phenstatin. The deprotected benzophenone phenstatin is a potent cancer cell line growth inhibitor. This inhibitor is structurally related to the known CBSI, podophyllotoxin [58]. Phenstatin is more stable than CA-4 in vivo as it cannot isomerize to an inactive *trans* configuration [59]. It mirrors CA-4’s biological mechanism of action and both phenstatin and its corresponding water-soluble prodrug salt demonstrate pronounced cytotoxicity against human cancer cell lines [60].

#### 4.2.3. IsoCA-4 and Selenium CA-4 Derivatives 

The 1,1-ethylene bridge (highlighted in blue, **14**, Figure 4) is regarded as a suitable bioisostere of the *Z* 1,2-ethylene bridge and has led to the development of potential *iso*CA-4 analogues. *Iso*CA-4 (**14**, Figure 4), with a 1,1-diarylethylene scaffold, is more active than phenstatin and colchicine (IC_50_ = 2–5 nM across a panel of cell lines) which are comparable values to CA-4 [61]. A diacrylonitrile derivative CC-5079 (**15**, Figure 4) had an IC_50_ of 3.0 nM comparable to 2.3 nM of CA-4 in HCT116 colorectal cancer cells [61]. During a screening program, it was found that 15 had potent antiproliferative and anti-tubulin effects on human umbilical vein endothelial cells (HUVECs) as an inhibitor of tubulin polymerisation, was associated with G2/M phase arrest and was active against MDR cell lines. It significantly inhibited growth of in vivo xenograft colorectal mouse tumour models [62]. Further to this work it was found that 15 inhibited angiogenesis in vitro and in vivo, potently inhibiting microvessel formation [63]. No further clinical development has occurred with these compounds. 

Few selenium containing CA-4 analogues are known in the literature. Those studied include selenium atoms as spacer groups between CA-4’s A and B aryl rings. Antiproliferative successes are good with similar anti-cancer potencies to CA-4 [64,65,66,67]. Pang et al. have prepared a series of organoselenium compounds obtained by introducing a methyl(phenyl) selane to *iso*CA-4 and phenstatin. Most of these compounds exhibit excellent and potent antiproliferative activity against human cancer cell lines. The most successful compound **16** (Figure 4) demonstrated low nanomolar potency with IC_50_ values of 3.9 nM in A549 cells, 2.2 nM in MDA-MB-231 cells and 3 nM in HEPG2 cell lines. Compound **16** also exerted successful antiproliferative activity against cisplatin resistant cell line A549/CDDP and doxorubicin resistant cell line HEPG2/DOX. The phosphate salt **17** (Figure 4) was tested against A549 xenograft tumours, resulting in an inhibitory rate of 72.9% compared to 47.6% of CA-4P and 52.2% of *iso*CA-4P, without apparent toxicity. Mechanistic studies demonstrate that the introduction of the selenium atom to *iso*CA-4 elevated its anti-cancer properties while retaining tubulin polymerisation inhibition, cell cycle arrest and apoptosis induction [68].

### 4.3. New Combretastatin A-4 Analogues Reported between 2017 and 2019

#### 4.3.1. New Combretastatin A-4 Analogues of 2017

##### β-Lactam *Cis* Restricted Analogues 

Our group has extensively reported a series of antiproliferative, tubulin-binding β-lactam compounds—the ‘combretazets’, which have greater tubulin depolymerization potency with respect to CA-4 [69]. Prior to our research, it was known that β-lactam-containing compounds possessed antiproliferative activity [70,71]. The introduction of the rigid β-lactam bridge (highlighted in blue, **18**, Figure 5) scaffold allows similar structural arrangement between CA-4’s two aromatic rings, overcoming double bond *cis*/*trans* isomerization by substituting the ethylene bridge for a 1,4-diaryl-2-azetidinone ring [69,72,73,74]. The rigid ring scaffold permits the similar spatial arrangement between the two aromatic rings as observed in non-planar *cis* conformations of CA-4.

These β-lactam bridge analogues of CA-4 have demonstrated significant tubulin depolymerising effects in human breast cancer cell lines [69,72,73,74,75]. Malebari et al. expanded the previous work of Greene et al. [69] on compounds **18** and **19** (Figure 5), which had IC_50_ values of 38 and 19 nM respectively in MCF-7 cells. Compounds with B ring *meta*-hydroxyl substituents, such as **18**, were far less active in combretastatin refractory HT-29 (human colorectal adenocarcinoma colon cancer) cells. [76]. Substitution or deletion of the B ring *meta*-hydroxyl group at the 4-position of the β-lactam ring to produce compounds such as **20** (Figure 5) led to potent activity in HT-29 cells. **20** had an IC_50_ value of 12 nM in HT-29 cells, contrasting with CA-4’s micromolar IC_50_ of 4.17 μM. Formation of a glucuronide metabolite is the main metabolic pathway by which CA-4 is cleared [77] and it is likely that glucuronidation is a method of intrinsic drug resistance in adenocarcinoma cells such as HT-29 cells [78]. Deletion of the B ring hydroxyl group slows glucuronidation by uridine 5-diphosphoglucuronosyl transferase enzymes (UGTs) in HT-29 cells and contributes to the potent activity of these β-lactams.

Analogues with thioether moieties in Ring B such as **21** and **22** (Figure 5) had potent antiproliferative activity in three cancer cell lines—MCF-7, HL-60 and HT-29. Interestingly the thioether β-lactams demonstrated excellent effects in HT-29 cells (IC_50_ values of 15 and 56 nM respectively). This demonstrates the ability of a thioether substituent to slow or avoid UGT-induced metabolism in this series of compounds, positioning them as lead compounds for resistant colon cancers. These compounds disrupt microtubule structure in both MCF-7 and HT-29 cells by inducing G_2_/M phase arrest and apoptosis [76]. Hence, CA-4’s B ring may provide fertile grounds for activity optimization of novel analogues for the treatment of MDR colon cancers. These compounds show promise for further clinical development having introduced two major modifications to prevent inactivation; firstly via isomerisation (*cis* restriction by replacement of the double bond with a β-lactam ring) and secondly via metabolism by glucuronidation (deletion of B ring *meta* hydroxyl group). 

Compounds **22**–**24** showed extremely high nanomolar potency in MCF-7 and HL-60 cell lines. IC_5o_ values for **22**, **23**, and **24** in MCF-7 cells were 4 nM, 4 nM, and 3 nM respectively. This emphasises the importance of the *meta*-hydroxyl for optimum activity in breast cancer cells, a large contrast to the Ring B’s detrimental role in terms of anti-cancer activity towards colon cancers. 

##### β-Lactam CA-4 Analogues with an Azide Substituted B Ring

β-Lactam **25** (Figure 5) was the most active of a series of azide derivatives synthesized by Fu and co-workers as an orally active CA-4 azide derivative. It acts as a tubulin polymerization inhibitor and antimitotic agent with similar biological effects to the aforementioned β-lactam derivatives (G_2_/M phase cell cycle arrest and apoptosis). Fu et al. describe replacing a phenyl at the C3 β-lactam ring position with a thiofuran sulphur-containing moiety to increase antiproliferative potency. The position of the azide is important for antiproliferative potency. Moving the N_3_ from *para* to *meta* position, as in **25**, greatly improved potency with IC_50_ values ranging from 0.106–0.507 μM compared to values of 0.154–4.203 μM in MGC-803, MCF-7 and A549 cells. Replacement of the azide structure with bulky groups such as 1,2,3-triazoles and long carbon chains had detrimental effects on antiproliferative potency. This demonstrated the importance of small substituent groups on CA-4’s B ring as large groups appear unfavourable for antiproliferative efficacy. In a xenograft model of MGC-803 tumour cells, **25** caused significant suppression of tumour growth without apparent toxicity and with comparative efficacy to CA-4P. This data suggests that **25** may serve as a new candidate for further investigation [79].

#### 4.3.2. New Combretastatin A-4 Analogues of 2018

##### Dihydronapthalene Derivatives 

Maguire and co-workers utilized the dihydronapthalene molecular scaffold (highlighted in blue, Figure 6) to synthesise two CA-4 derivatives KGP03 (**26**, Figure 6) and KG413 (**27**, Figure 6) as promising inhibitors of tubulin polymerisation. Their work was based on prior investigation of Oxi8006 (**28**, Figure 6), an indole derivative [80] and phenstatin (13) [58,81]. Compounds **26** and **27** were found to have IC_50_ values for the inhibition of tubulin polymerisation of 0.46 and 0.85 μM, akin to CA-4 (IC_50_ = 1.2 μM). This prompted evaluation of corresponding water-soluble prodrugs KGP04 (**29**) and KGP152 (**30**, Figure 6) in in vivo models of human cancers using murine models of MDA-MD-231 breast orthotopic tumours utilising bioluminescence imaging (BLI). Further, **30** induced vascular shutdown at 4 h, with a decrease of 99% relative to control in light emission, which had only recovered to 81% by 24 h after administration. Similarly, KGPO4 (**29**) was administered in a rat model of human lung cancer with a human A549 tumour xenograft. Vascular shutdown as measured using Doppler Colour ultrasound was seen as early as 2 h following administration. The dihydronaphthalene analogues appear to be promising candidates for further development as VDAs [82]. 

##### Quinazolinone Derivatives

Wolfgang et al. have investigated 3,4-dihydroquinazolin-2(1H)-one (highlighted in blue, Figure 6) derivatives as CBSIs. Through replacement of steroidal motifs of 2-methoxyestradiol, while maintaining essential pharmacophore regions, they synthesized a derivative with nanomolar potency (**31**, Figure 6) (GI_50_ of 50 nM in DU-145 and MDA-MB-231 cells). Tetrahydroisolquinoline moieties were used to mimic A and B steroidal rings, while the N-2 position is tethered to a benzyl group mimicking the steroidal D ring. This series sees the introduction of methoxyl benzyls at this N-2 position with the aim of producing a steric clash between the benzyl ring and N-2 position. A steric clash places a H-bond acceptor group near an optimal binding target at the CBS. Indeed, co-crystallization of **31** with the αβ-tubulin heterodimer demonstrates that **31** binds more deeply than colchicine itself. The sulfamate group is shown to be a mediator of this affinity with β-tubulin, the first example of a sulfamate ester bound to tubulin. Future studies will potentially involve evaluation of this compound in animal models [83].

##### Photo-Responsive Azo CA-4 Analogues

Rastogi et al. [84] have adopted an interesting approach to improve the potency and stability of CA-4 as a cytotoxic agent, recognising the promise of a photo-responsive approach in the area of MTAs. They note that while non-isomerizable analogues of CA-4 may display more potent anti-cancer activity, there still remains issues with drug specificity. This problem persists as the molecules are non-selective towards normal and cancer cells. To address this issue of tissue selectivity they have designed a series of potent photo-responsive analogues of CA-4 in which the double bond is replaced by an azobenzene structure (highlighted in blue, Figure 6). This has previously been investigated by two other groups [85,86]. Using UV light in ranges of 395–400 nm, it is possible to convert an inactive *trans* isomer to *cis* bioactive derivatives in a selective manner. Photo-pharmacology is a potentially powerful tool to selectively target chemotherapy towards cancer cells only. Biological functions can be controlled by applying light to localized areas, achieving in vivo accuracy to overcome poor specificity in drug delivery [87,88,89].

The focus of the work of Rastogi et al. was to modify the B ring in order to improve potency of these analogues. *Trans* compound **32** (Figure 6) had IC_50_ values of 60 and 110 μM in HeLa and H157 cells in the dark respectively. Upon irradiation to the *cis* compound **33** using 390-400 nm wavelengths of UV light, the antiproliferative activity improved 550-fold. This technique shows promise for site specific activation of chemotherapy, a very interesting and emerging approach. The photoresponsiveness of these analogues is easily seen using UV-spectroscopy, monitoried by n-π* and π-π* transitions [82]. 

##### Replacement of the Trimethoxyphenyl Moiety

Wang et al. recently investigated the replacement of the essential 3,4,5-trimethoxy moiety of CA-4 for improvement of antiproliferative and tubulin binding activity. Modifying this portion of CBSIs is typically unsuccessful in improving antiproliferative potency. Crystallography work of analogues by this group demonstrated that only one methoxy moiety was involved in an interaction with Cys-241 of β-tubulin. Hence, the other two methoxys could be modified to improve antiproliferative activity. This resulted in **33** (Figure 6), the most potent of the series, containing a unique 3-methoxybenzo-[4,5]-dioxene moiety (highlighted in blue, Figure 6. It demonstrates IC_50_ values ranging from 1.1–3.3 nM in several melanoma cell lines. Co-crystallisation with tubulin has confirmed its direct binding to the CBS, demonstrating that the A ring of CA-4 may indeed be further optimized to improve potency [90].

#### 4.3.3. New Combretastatin A-4 Analogues of 2019

##### ABI-231 Analogues

Chen et al. have recently reported novel analogues of ABI-231 (Figure 6), one of their most potent orally bioavailable CBSIs with a bicyclic heterocycle pharmacophore, central imidazole and carbonyl linker (highlighted in blue, Figure 6) [91,92]. Amongst all of this group’s scaffolds, ABI-231 is by far the most potent and successful antiproliferative compound having an average IC_50_ of 5.2 nM in terms of antiproliferative activity across a large panel of cell lines. Additionally, ABI-231 inhibits the expression of mutant tubulin isotypes inclusive of βIII and βIV tubulin thus restoring cellular susceptibility towards the action of typical MTAs, supporting its use to surmount MDR [93]. The authors have extensively investigated SAR of the 3-indole moiety to provide a rational for molecular interactions at the CBS. They identified two superior analogues **34** and **35** (Figure 6). X-ray crystallography confirms unique and direct binding to the CBS demonstrating superior interactions to β-tubulin to explain increased antiproliferative activity. Both analogues successfully disrupt tubulin polymerisation, promote microtubule fragmentation and inhibit cancer cell migration. Compound **34**, the 4-methyl analogue of ABI-231, demonstrated the best inhibitory effects with an IC_50_ range from 1.7 to 3.2 nM in A375, MI4 and WMI64 cells which is 3-fold more potent than parent ABI-231. Compound 35 was synthesised based on the hypothesis that an indolyl rotation may increase binding affinity at the CBS and augment antiproliferative activity [94]. The 4-indolyl analogue showed approximately two-fold improvement in activity compared to ABI-231 with IC_50_ values ranging from 1.6–3.7 nM in the aforementioned cell lines compared to 5.6–8.1 nM for parent compound ABI-231. These compounds are active against MDR cell lines, including those resistant to paclitaxel, thus circumventing P-gp-mediated resistance. Compound 35 demonstrated high potency in the inhibition of tumour growth in a prostate paclitaxel resistant mouse xenograft model, suggesting that it may be a promising compound for certain MDR cancer types [95].

##### 3-Vinyl Substituted β-Lactams 

Our group reported a series of 3-vinyl β-lactams with potent anti-tubulin and antiproliferative effects on breast cancer cells. Compound **36** (Figure 7) was the most potent of the series with a mean GI_50_ value of 23 nM when tested across the National Cancer Institute panel of cell lines. In particular, this compound showed potent activity in the MCF-7 breast cancer cell line with an IC_50_ of 8 nM, comparable to CA-4. Compound **36** inhibited tubulin polymerisation 8.7-fold at a concentration of 10 μM. This study demonstrated that a hydrophobic substituent at C-3 of the β-lactam ring may enhance biological activity. Preliminary docking studies revealed that the vinyl substituent sits in the CBS. These results are promising for further development of the β-lactam structures as more stable analogues of CA-4 [72].

##### Pyrazole Analogues 

Romagnoli et al. previously published a series of potent analogues in which CA-4’s ethylene bridge is incorporated into 3,4-diaryl substituted *1H*-pyrazoles (highlighted in blue, Figure 7) and tetrazoles [96,97]. This work was based on the work of Medarde and co-workers from 2005, who described 3,4-disubstitued pyrazole derivatives of CA-4. Their most potent inhibitor of tubulin polymerisation, compound **37** (Figure 7), had an IC_50_ value of 33 μM compared to 3 μM of CA-4 [98].

Despite CA-4’s potency in many cancer cell lines, it is ineffective against HT-29 colon cancer cells. Two analogues **38** and **39** (Figure 7) with a 4-ethoxyphenyl at either the C-4 position (**38**) or C-3 position (**39**) of the *1H*-pyrazole ring show great promise in these cell lines. While previous work involving the use of a pyrazole *cis* restricted CA-4 analogue resulted in compound **37** demonstrating potencies of 3.3–28.4 nM across a range of cancer cell lines, by comparison **39** demonstrated superior in vitro inhibitory effects on tumour cell proliferation across all cell lines with superior IC_50_ values of 0.06–0.7 nM. Interestingly **39** was most effective in HT-29 cells, a cell line with known resistance to CA-4 due to the expression of UGTs. As mentioned above, the overexpression of UGTs leads to glucuronidation of CA-4’s *meta* hydroxyl group of Ring B [76].

As **38** also demonstrated subnanomolar IC_50_ values, it was concluded that the position of 3,4,5-trimethoxyphenyl at C-3 or C-4 of the 1*H*-pyrazole had little effect on the antiproliferative activity. On the contrary, the position of the aromatic rings at the 3 or 4 position of the 1*H*-pyrazole system affected the potency with **39** being 4–6 times more potent than its isomeric counterpart **38** with the exception of HeLa cells. Both **38** and **39** successfully inhibited tubulin polymerisation, indicating that a pyrazole can substitute successfully for CA-4’s double bond. Both compounds show cytotoxicity against CEM^Vbl−100^ cell line which is an MDR cell line overexpressing the P-gp efflux transporter [99]. This indicates that these derivatives are not substrates of P-gp. Compound **39** also prevented cell migration in an MDA-MD-231 breast cancer cell model. Compounds **38** and **39** were evaluated for cytotoxic potential in peripheral blood flow lymphocytes in healthy donor cells and non-cancerous human astrocytes. Both compounds had IC_50_ values greater than 10 μM, together suggesting low toxicity in normal cells in comparison to tumour cells. This preliminary data is particularly promising as neurotoxicity is a major limiting factor in the clinical use of antimitotic agents. Lead compound **39** was assessed in a mouse allograft tumour model using E0771 murine breast cancer cells and preliminary results show reduction in tumour burden comparable to CA-4P, with no toxic effects or weight loss [97].

A series of pyrazole *cis*-restricted CBSIs was also recently described, including novel analogue **40** (Figure 7) which introduces a methyl at the N-1 position of the pyrazole ring. Compound **40** is an analogue from a series of 1,4-substituted-3-(3,4,5-trimethoxyphenyl)-5-aminopyrazole analogues of CA-4. **40** had the best antiproliferative activity with an IC_50_ comparable to CA-4 of 17 nM and 31 nM in ovarian cancer cells and triple-negative breast cancer cells respectively. Larger bulky groups replacing the N-1 methyl were not tolerated. Crucial SAR contributions were the 4-ethoxy and 4-methylthiol on the B ring, with 3-methoxy groups reducing activity. This group demonstrated that by placing the A ring at the C4 position instead of the C3 position, only a minor effect in activity was observed. Pyrazole **40** was administered to mice containing 200 cm^3^ human ovarian adenocarcinoma xenograft model tumour. It resulted in reduction of tumour growth compared to vehicle by 62.8% whilst being well tolerated. Compound **40** exhibits typical cellular effects of CBSIs, including G_2_/M phase arrest, apoptosis, the inhibition of cancer cell migration and disordered spindle formation/microtubule polymerization [100].

##### Piperazine Conjugates

Our group has recently described a series of CBSIs which consists of piperazine-substituted derivatives (highlighted in blue, Figure 7) of CA-4 formed via two-step Perkin microwave synthesis and amine conjugation. Piperazine may improve water solubility of drug molecules due to the presence of a heteroatom and in this case was chosen to prevent unwanted *cis* to *trans* isomerisation of CA-4. Three compounds **41**, **42** and **43** (Figure 7) had submicromolar IC_50_ values in MCF-7 cells during preliminary in vitro screening. Docking studies indicate that the trimethoxy A rings overlay with with those of DAMA-colchicine, whilst the B ring of the piperazine conjugates align with the the C ring of DAMA-colchicine. Toxicity of the compounds is minimal and water-soluble conjugates will be developed for the potential treatment of triple-negative breast cancer [101].

##### Oxazole-Bridged Analogues

Schmitt et al. published a series of novel pleiotropic CA-4 derivatives which includes both CBSIs and inhibitors of histone deacetylase enzymes (HDACs). It is known that HDACs are overexpressed in various solid tumours [102]. Sole HDAC inhibitors have shown shortcomings in treatment of solid tumours. In order to overcome such drawbacks, inhibitors termed designed multiple ligands with dual or multimodal actions are sought after [103]. HDAC inhibitors show synergistic effects in combination with tubulin-binding anti-cancer drugs. This group describe a series of tubulin-targeting oxazole-bridged derivatives with hydroxamate appendages (highlighted in blue, **44**, Figure 7) [104]. These compounds were potent antiproliferative compounds against Ea.Hy926 cells (endothelial hybrid cells) with IC_5o_ values ranging from 1.2 to 410 nM, with selectivity over non-malignant human dermal fibroblasts (HDFa cells; IC_50_ 23.9 to >100 μM). Antiproliferative activity was strongest with decreasing linker length. Compound **44** (Figure 7) containing a 3-carbon linker completely inhibited tubulin polymerization at a concentration of 10 μM. In contrast, compound **45** (Figure 7) containing a 5-carbon linker, showed no inhibition whatsoever, supporting the theory that cytotoxic effects of the compounds decrease with increasing carbon chain length via absence of tubulin inhibition. Compound **45** in contrast was the most potent HDAC6 inhibitor demonstrating the need to find a balance between the two activities. Compound **44** is a promising drug candidate with antiproliferative, cell cycle arresting, microtubule-destabilizing effects in addition to HDAC inhibition. In vivo investigations are underway [104].

##### Quinoline and Indole Derivatives of *iso*CA-4

Li et al. synthesised **46** and **47** (Figure 7), where CA-4’s 3,4,5-trimethoxyphenyl and *iso*CA-4’s isovanillin moieties are replaced with a quinoline moiety and indole moiety respectively (highlighted in blue, **46**, Figure 7). The use of the quinoline moiety is based on work by Khelifi et al. in which a quinoline structure was predicted by docking studies to form a hydrogen bond with Cys-241 residue of the CBS through the *N*-1 atom of the quinoline ring. Quinoline **48** (Figure 7) displayed nano- and sub-nanomolar levels of cytotoxicity against five cancer cell lines and inhibited tubulin polymerization with micromolar IC_50_ values [105].

The adoption of the indole moiety in place of the isovanillin structure resulted from studies demonstrating its occurrence in CBSI’s as a replacement for isoCA-4’s isovanillin ring moiety [106,107,108]. Compounds **46** and **47** showed most potent activity against leukemic K562 cells and five other cancer cell lines with IC_50_ values ranging from 5 to 11 nM, comparable to those of CA-4. In vivo anti-tumour activity was evaluated for both leads in a liver cancer xenograft mouse model. Interestingly **46** and **47** reduced tumour weight by 63.7% and 57.3% respectively without apparent toxicity. Both compounds were more potent than CA-4 which inhibited tumour growth by 51%. Collectively these results highlight the potential in further development of **46** and **47** as tubulin-targeting anti-cancer drugs [109].

##### Heterocyclic *iso*CA-4 Derivatives 

Naret et al. have undertaken the challenge of optimizing *iso*CA-4 analogues with heterocycles replacing both traditional A and B rings, where the 3,4,5-trimethoxyphenyl A ring is replaced by a pyrimidine-dionyl, quinolinyl or a quinazolinyl while the B ring is substituted by a carbazolyl or indolyl group. Compound **49** (Figure 8) resulted with highest potency with a quinaldine moiety as ring A and an *N*-methylcarbazole (highlighted in blue, Figure 8) in place of ring B. Compound **49** was more active than *iso*CA-4 against A549, U87-MG and HUVEC cells and it was 67-fold more potent than CA-4 in lung adenocarcinoma epithelial cells (A549). When tested in a range of MDR cells lines, **49** was up to 27 times better than *iso*CA-4 and CA-4, including against HT-29 cells (IC_50_ of 3 nM compared to 265 μM and >8000 μM for *iso*CA-4 and CA-4 respectively) [110]. It has a high logP value of 5.26 and therefore potentially could partition through the blood-brain barrier. Combined with activity against human glioblastoma cell lines (U87-MG, IC_50_ = 1.9nM), **49** could potentially serve as a candidate for the treatment of glioblastoma, an exciting prospect for the novel compound [108].

##### Novel Benzosuberene Analogues 

Niu et al. replaced the double bond of CA-4 with a benzosuberene (highlighted in blue, Figure 8) ring structure to yield compounds **50** and **51** (Figure 8), with an IC_50_ of 6.9 nM in the ovarian cancer cell line SK-OV-3. The phosphate prodrug **51** showed dose-dependent vascular shutdown comparable to CA-4 [111].

##### 1,2,4-Triazole-3-Carboxamide Derivatives 

A new study has recently been published using a 1,2,4-triazole-3-carboxamide scaffold (highlighted in blue, Figure 8) to replace the stilbene double bond of CA-4. Compound **52** (Figure 8) was the most potent analogue of the synthesized series, with an IC_50_ of 4 nM in HL-60 cells, an excellent example of the introduction of fluorine leading to enhanced cytotoxicity [112].

## 5. CBSIs Derived from Sources Other Than Combretastatins

### 5.1. Podophyllotoxin and Analogues 

Podophyllotoxin (**53**, Figure 8) is a naturally occurring MTA. It binds to tubulin causing cells to accumulate in metaphase in a similar fashion to colchicine. It is the major chemical constituent of podophyllin, the alcoholic extract of the *Podophyllum* plant rhizome. These plants have long been used by indigenous populations of North America and in the Himalayas. Podophyllin was first reported in 1946 to demonstrate toxic effects against mitotic cells in similar manner to classical MTAs such as colchicine [113]; this activity was later attributed to the active component podophyllotoxin inhibiting assembly at the mitotic spindle. Despite its original promising ability to inhibit mice tumour growth and potential for translation towards treatment of human malignant tumours [114], unacceptable gastrointestinal toxicity limits its use as a chemotherapeutic agent [115]. 

This led to a body of work involving podophyllotoxin’s optimization, culminating in semi-synthetic derivatives etoposide (**54**, Figure 8) and teniposide (**55**, Figure 8) by Sandoz Ltd. who studied glycosides of the podophllotoxin aglycon structure. Upon condensation of benzaldehyde with the podophyllum glycoside fraction, a highly active antiproliferative agent was isolated and identified as 4’-*O*-demethyl-epipodophyllotoxin benzylidene- β-D-glucoside (DEPBG) (**56**, Figure 8). Condensing 4’-O-demethyl-epipodophyllotoxin glucoside led to discoveries of etoposide (**54**) and teniposide (**55**) [116]. Currently, both are licensed for the treatment of malignant tumours: etoposide for various cancers in the EU and teniposide (VUMON^®^) for the treatment of refractory childhood acute lymphoblastic leukaemia licensed by the FDA in the USA [117]. The mode of action of these compounds is somewhat distinct from podophyllotoxin. These compounds are known to inhibit assembly at the mitotic spindle and induce cell cycle arrest at mitosis via binding to the CBS, in addition to acting as a DNA topoisomerase II poison [118]. 

Continuous use of these agents remains an issue; they lead to several adverse effects including myelosuppression, acquired drug resistance, non-specific cytotoxicity and the development of secondary leukaemia [119]. The impressive anti-tumour potency and clinical efficacy of etoposide and teniposide has prompted extensive SAR studies and molecular modifications of the podophyllotoxin prototype to construct several synthetic analogues—some of which have already undergone clinical trials for various cancers but none are CSBIs. Etopophos (**57**, Figure 8) is a water-soluble phosphate (highlighted in blue, Figure 8) prodrug which has been developed by Bristol Myers and appears less toxic while more active than its parent compound [120]. It has superseded etoposide for routine clinical use [121].

### 5.2. Chalcones

#### 5.2.1. Millepachine and Derivatives 

The chalcone scaffold is the basis for several potent CBSIs. Chalcone combretastatin derivatives synthesised by Ducki et al. [56,57] (Figure 2) were previously discussed in Section 4.2.1. More recently, work has been carried out based on a naturally isolated chalcone named millepachine. Chalcones (**58**, Figure 9) are naturally occurring polyketide compounds that have been isolated from various plants. Chemically they consist of two aromatic phenyl rings joined by an αβ-unsaturated enone system. They display various biological activities and in the last 25 years have emerged as an interesting class of potential anti-cancer agents. Chalcones were first recognised as cytotoxic agents in the 1970’s. A series of mono and dichloro nitrochalcones was described with cytotoxic properties [122]. Chalcones were first discovered to act as potent antimitotic agents in 1990, by Edwards et al. Their compound MDL-27048 (**59**, Figure 9) was effective at 4 nM in HeLa cells as a cytotoxic agent. It inhibited tubulin polymerisation in vitro at a lower concentration (1 μM) compared to colchicine (6 μM). Compound **59** bound rapidly with high affinity and reversibly to the tubulin heterodimer at the CBS, constituting a powerful and specific antimitotic agent [123]. Ducki et al. have comprehensively outlined chalcones dating up to 2007 in development as promising anti-cancer agents [57]. This review will discuss more recent publications. 

*Millettia pachycarpa* Benth (Leguminosae) is a flavonoid-rich traditional Chinese medicine which has been used as a blood tonic in anti-helminthic and as an anti-cancer preparation called ‘Jixuteteng’ in Chinese medicine for many years. Millepachine (**60**, Figure 9) is a relatively new chalcone with a 2,2-dimethylbenzopyran motif (highlighted in blue, Figure 9), isolated from the seeds of *M. pachycarpa.* Chalcone **60** has shown significant antiproliferative effects against several cancer cell lines. It has been evaluated in HepG2 tumour-bearing in vivo xenograft mice models at three concentrations and resulted in inhibition of tumour growth by 27%, 42% and 66% respectively. This compared with a 48% inhibition in mice treated solely with the control doxorubicin [124]. Poor solubility limited further dose increases and modification of **60** was necessary. Until recently **60** was characterised as a cytotoxic antimitotic agent but the precise cellular targets were unknown. Cao et al. synthesised derivatives of millepachine which, despite targeting the CBS, appeared to stabilize tubulin in a similar fashion to docetaxel, opposing colchicine’s destabilizing effect. This stabilization contradicted the paradigm of typical CBSIs as tubulin destabilizing agents [125].

Yang et al. have recently carried out biochemical and cellular experimentation, revealing through X-ray co-crystal structures of tubulin-millepachine derivatives that millepachine and derivatives were CBSIs [126]. Ring B modifications increased antiproliferative properties for millepachine derivatives. The introduction of an amino group to the B ring (**61**, Figure 9) increased potency by up to 253-fold in a panel of cancer cell lines from IC_50_ ranges of 1.51–4.0 μM for parent compound millepachine (**60**) to an average IC_50_ of 61 to 8–26 nM for amino substituted **61**. Chalcone **61** was more potent than both paclitaxel and colchicine against C26, MCF-7, HEPG2 and ES-2 cell lines. The addition of the amino at the 3-position of the B ring caused a significant increase in antiproliferative activity whereas substitution of this same amino group resulted in loss of antiproliferative activity. Molecular modelling of **61** revealed that the B ring binds deep within the CBS. The amino moiety provides an extra hydrogen bonding interaction with Val238 and stabilizes **61**’s interaction. This explains the greater potency of **61** in comparison to **60**. Preliminary data suggests that **61** has a much lower resistance factor than paclitaxel, cisplatin and adriamycin, indicating that **60** and derivatives are unaffected by P-gp and mutant β111 tubulin isotype expression and may be useful in treatment of refractory tumours. Compound **61** was also found to act as a potent VDA, inhibiting capillary-like tube formation in endothelial cells in concentrations as low as 50 nM. The hydrochloride salt of **61** shows promising results in in vivo models, reducing tumour growth without causing significant weight loss. In addition, **61**’s hydrochloride salt appears to have good oral bioavailability up to 47% [124].

Emerging research suggests that another potential mechanism of overcoming drug resistance is the design of irreversible covalent inhibitors at β-tubulin. Yang et al. found SKLB028 (**62**, Figure 9) bound irreversibly to tubulin [126]. Cells failed to recover from cell cycle arrest up to 72 h after treatment, indicating that covalent bonding had occured.Designing anti-cancer compounds which covalently bind to their target is a well-known approach to overcome drug resistance [127,128]. Unlike the scenario in which binding is rapid, non-covalent and reversible whereby tumour cells acquire resistance by reducing target affinity or enhancing drug efflux through P-gp, MRP1 and MRP2 overexpression, resistant tumour cells cannot escape the effects of irreversible covalent binding after initial exposure to the compound.

X-ray crystal structures of free millepachine compared to a co-crystal of **61** in complex with tubulin have assisted the lead optimisation process to maximise the potency of these promising derivatives. Unbound **61** adopts an s-*cis* conformation but binds into the CBS in s-*trans* configuration. Wang et al. hypothesised that an *s*-*trans* conformation would prove more potent. An α-methyl group was introduced to create α-M-SKLB050 (**63**, Figure 9) and α-M-SKLB028 (**64**, Figure 9). This structural change is thought to mediate *s-trans* conformations via steric repulsion that would exist between the methyl introduced and the A-ring in an s-*cis* conformation. Chalcone **63** induced cell cycle arrest at an average concentration of 5 nM versus 24.8 nM for **61** without the additional methyl group, across five cells lines. Similarly, **64** had a promising average IC_50_ of 21 nM versus 131.5 nM for **62** [126]. These compounds have great potential as covalent inhibitors to target cancers which cannot be treated by standard, reversibly-binding MTAs.

#### 5.2.2. New Quinolone Chalcones: CBSIs and Inhibitors of MRP1 Function 

Screening of a library of compounds led to the identification of two compounds, CTR-17 (**65**, Figure 10) and CTR-20 (**66**, Figure 10), as promising lead chalcone derivatives selective for three breast cancer cell lines compared to healthy cells. Despite striking differences in structure when compared to colchicine, they appear to occupy the CBS. Compound **65** and **66** were effective in *MDR1* and *MRP1*-overexpressing MDR cells, in comparison to colchicine which was ineffective. Compound **66** is an inhibitor of the MRP1 efflux pump which contributes to its successful activity in MDR cell lines. 

Both compounds were extremely effective in mouse models of MDA-MB-231 triple-negative breast cancer. The treatment of mice with either **65** or **66** (30mg/kg twice a week for 30 days) successfully suppressed tumour growth. Early pre-clinical animal data is promising. Half the standard dose of paclitaxel combined with **66** proved more efficacious in animal tumours than either drug alone as a monotherapy, suggesting that this combination regimen holds potential to achieve better therapeutic results at a lower dose against MDR tumours. These lead compounds could be an excellent solution as adjunctive therapy to overcome paclitaxel’s toxicity and MDR shortcomings [129].

#### 5.2.3. Quinoline-Chalcone Derivatives of 2019

Li et al. have utilised a quinoline moiety (highlighted in blue, Figure 10) as a surrogate of CA-4’s 3,4,5-trimethoxyphenyl ring with the aim of increasing aqueous solubility for clinical use, as the development of many CBSI’s in the past has been limited by poor aqueous solubility [129,130]. The most potent compound of the series **67** (Figure 10) was effective in a range of cell lines with IC_50_ values of 9 nM, in K562 human cancer cells and 15 nM in three cell lines, HEPG2, HCT8 and MDA-MB-231. It inhibited tubulin polymerisation with an IC_50_ of 1.7 µM (CA-4: 2.5 μM), caused G_2_/M arrest, depolarized mitochondria and induced reactive oxidative stress in K562 cells. Compound **67** successfully induced apoptosis by interfering with the expression of apoptotic proteins, increasing the expression of pro-apoptotic Bad and Bax proteins and downregulating the expression of anti-apoptotic Bcl-2 and Bcl-xl proteins [131].

Compound **67** appeared to be 16 more times more soluble in aqueous media than CA-4 with concentrations of 16 μg/mL attainable, likely attributable to the quinoline moiety. Additionally, the hydrochloride salt of parent compound **67** had a solubility of 1 mg/mL which is a promising starting point for modification of its physiochemical properties for use in clinic. Parent compound **67** had excellent potency in mouse liver cancer xenograft models, suppressing tumour growth by 65% with no observable toxicity. The hydrochloride salt outperformed **67** with tumour growth suppression of 69% using an identical treatment regimen. The group treated with **67** also had a much lower tumour microvessel density when compared to control, confirming the anti-vascular activity of these compounds. Compound **67** also inhibited migration and invasion of MDA-MB-231 cells, indicating potent anti-metastatic activity and potential for the treatment of resistant and aggressive triple-negative breast cancer subtypes. Collectively, this data suggests that **67** has potential for further development as an effective chemotherapeutic agent [132].

### 5.3. Curacin A 

Curacin A (**68**, Figure 10) is the name given to a complex ketopeptide originally isolated from cyanobacterium *Lynbya majuscule*. It inhibits mitosis, tubulin polymerisation and colchicine-binding in a competitive manner through strong non-covalent binding to the CBS [132]. The structure of curacin A is unique amongst CBSIs as it lacks the recurrent aromatic moieties. Instead it bears two simple conjugated olfenic bonds [133]. Despite curacin A’s excellent nanomolar antiproliferative potency in human cancer cell lines in vitro, its clinical development is hindered by high lipophilicity and therefore poor water solubility. In vivo the molecule is essentially inactive due to its water insolubility and instability [134]. Attempts have been made to synthesise superior derivatives suitable for clinical use. None appear to have advanced for further pre-clinical or clinical development at present. 

### 5.4. Imidazo[4,5]pyridine DJ95 (DJ101)

Wei and co-workers have previously synthesised multiple series of CBSIs, with IC_50_ values in the nanomolar range, which are highly potent against MDR cell lines. These analogues are not substrates of P-gp and thus effectively overcome P-gp-mediated resistance. The arylbenzylimidazole lead analogue ARB1-111 (**69**, Figure 10) had an average IC_50_ of 3.8 nM and was particularly potent in melanoma and prostate cell lines [91]. Amongst the 4-substituted methoxybenzoylarylthiazole (SMART) series, compound **70** (Figure 10) was the most potent but limited by poor aqueous solubility. This was overcome by the addition of polar groups to create the orally bioavailable phenylaminothiazole derivatives with an amino linkage between rings A and B—of which, **71** (Figure 10) was the most potent derivative. Despite excellent activities in xenograft models, the ketone moiety present in all compounds was a metabolically labile site. To block apparent in vivo ketone reduction, the carbonyl link was modified. A new D ring was introduced between the A and B rings to mimic the carbonyl group structure. A potent new lead compound DJ95/DJ101 (**72**, Figure 10) was identified utilising an imidazo[4,5]pyridine-fused ring template (highlighted in blue, Figure 10). This compound showed increased potency relative to parent SMART compounds with an average IC_50_ of 5 nM in melanoma and prostate cancer cells [135].

High-resolution crystal structures of αβ-tubulin in complex with **72** confirm binding at the CBS, forming three hydrogen bonds with the tubulin heterodimer [136]. Tested against NCI-60 cell lines, GI_50_ values were less than 10 nM. In vivo xenograft models in A375 models of lung metastases in nude mice demonstrated 66% and 93% tumour growth inhibition at doses of 15 mg/kg and 30 mg/kg, with negligible signs of toxicity. It induced G_2_/M phase cell cycle arrest with an element of vascular disruption.

Compound **72** also showed a 6.2-fold decrease in lung metastasis after two weeks in an experimental model of lung metastasis in mice. Interestingly **72** appears to be most effective in resistant tumours. In vivo docetaxel and paclitaxel outperformed **72** in non-resistant cell lines, with tumour growth inhibition of 101% for paclitaxel-treated mice versus 79% in **72**-treated xenograft mice. Using identical dosing schedules and frequencies in a paclitaxel resistant cell line, tumour growth inhibition was reduced by 104% by **72** versus a modest 38% using docetaxel. Therefore, **72** represents an excellent lead candidate which is effective against a broad range of resistant metastatic melanomas. Pre-clinical evaluation by Kinsie et al. supports further development as a metabolically stable, novel treatment for resistant and metastatic melanoma and other cancers. Furthermore, recently published data in 2019 evaluating **72** in ABC-transporter overexpressing cell lines concluded that this compound overcomes MDR in cancer. As with many other agents discussed in this review, it may prove an effective alternative treatment for patients when other MTAs fail to show efficacy due to acquired MDR [137].

### 5.5. 2-Methoxyestradiol and ENMD1198

2-Methoxyestradiol (**73**, Figure 11) is an endogenous metabolite of estradiol and is known to destabilise microtubules and have anti-angiogenic effects [138]. Compound **73** is metabolised in vivo via conjugation at positions 3 and 17 (highlighted in blue, Figure 11) and oxidation at position 17 to render the compound inactive. To make 2-methoxyestradiol more metabolically stable for use as a VDA, a new lead compound and CBSI ENMD-1198 (**74**, Figure 11) (Table 1) was generated by chemical modification at the C-3 and C-17 positions [139]. Of a series of analogues, **74** showed most promise and was subsequently selected as the lead compound proceeding to phase I clinical trials in 2008, in patients with refractory solid tumours as a novel antimitotic agent, sponsored by CASI pharmaceuticals but has not yet made it to clinic. Little data has been released since 2010. Compound **74** is up to 6-fold more potent than parent compound at inhibiting endothelial cell proliferation, in addition to the potent inhibition of cell motility, tumour angiogenesis, chemotaxis and morphogenesis into capillary-like structures. It blocks vascular endothelial growth factor receptor 2 (VEGFR-2) expression in endothelial cells thus blocking endothelial cell processes involved in promoting angiogenesis. Thus, **74** is an innovative CBSI, with promising and potent anti-vascular activity [140].

### 5.6. BAL27862

BAL27862 (**75**, Figure 11) is a novel synthetic microtubule-targeting agent discovered via high intensity throughput optimization following high-throughput screening. It induces G_2_/M phase arrest, destabilizes microtubules and has a unique microtubule-severing activity while being a potent inhibitor of tumour cell growth. It holds remarkable potency against tumour cell lines resistant to clinically relevant MTAs such as the vinca alkaloids and taxanes. In vitro, **75** demonstrated potent antiproliferative activity with a median IC_50_ of 13.8 nM (range 5.4 nM to 25.2 nM) [141]. Despite the structural dissimilarity with colchicine, docking studies have shown that the benzimidazole and 3-amino oxadiazole moieties (highlighted in blue, Figure 11) superimpose well with C and B rings of colchicine. The cyanoethyl side chain and B ring of colchicine occupy similar binding pockets, while the aminophenyl (highlighted in blue, Figure 11) and A ring of colchicine are also in the same pocket; the amino phenyl contacts the H-7 helix of β tubulin only while colchicine’s A ring interacts with H-7 and H-8 helices [20]. It is also known as plinabulin and is now in clinical trials (Section 6.2.2 below).

### 5.7. Nitrobenzoate IMB5046

A novel MTA 2-(4-morpholinyl)-5-nitro-benzoic acid[4-(methylthio)phenyl]methyl ester (IMB5046, **76**, Figure 11) exerts the potent inhibition of multiple tumour cells lines with an IC_50_ range of 37–400 nM. It binds to the CBS even with the absence of the traditional 3,4,5-trimethoxyphenyl moiety. Compound **76** inhibited tubulin polymerisation with an IC_50_ of 2.97 μM. A limited proteolysis assay suggested **76** binds at the CBS. Molecular modelling studies indicated a slightly different binding mode in comparison to colchicine; hydrogen bonding to Lys254 in addition to hydrophobic interactions with Leu248 and Ala317 exclusively at the β-subunit. Compound **76**, in contrast to colchicine and other MTAs, overcame drug resistance in P-gp overexpressing KBv200 cells. Thus, **76** could act as a promising lead and novel scaffold targeting the CBS in a divergent manner when compared to other CBSIs discussed in this review [142,143].

### 5.8. Imidazole BZML

5-(3, 4, 5-Trimethoxybenzoyl)-4-methyl-2-(*p*-tolyl) imidazole (BZML, **77**, Figure 11), developed by Bai et al., is now in pre-clinical stages with strong cytotoxic activity and low nanomolar IC_50_ values across a family of human cancer cell lines inclusive of MDR subtypes, superior to those of CA-4. Despite striking deviations from colchicine’s structure, **77** is a competitive inhibitor of colchicine at the CBS. It was found to be an irreversible modulator of P-gp function in paclitaxel-resistant lung cancer cells by decreasing P-gp expression at the protein and mRNA levels. This characteristic explains its anti-MDR properties at least in part. BZML may yet again offer another novel strategy to solve issues of drug resistance [144].

### 5.9. Crolibulin (MX-58151)

MX-58151 (crolibulin, **78**, Figure 11), belonging to a novel series of 2-amino-4-(3-bromo-4,5-dimethoxy-phenyl)-3-cyano-4H-chromenes (chromene highlighted in blue, Figure 11), was identified during a cell-based high throughput screening assay by Gourdeau and co-workers [145]. It was identified as a tubulin destabilizer with potent in vitro cytotoxicity specific to cancer cells as a CBSI. Compound **78** was particularly active against paclitaxel-resistant human tumour cell lines with a GI_50_ value of 2.5 nM. Renamed as crolibulin, it has been developed by EpiCept Corp. in California (Section 6.2.4 below). Crolibulin is a potent VDA with capacity to inhibit capillary tube formation at 30 nM, comparable to CA-4’s value of 10 nM.

### 5.10. Novel Nicotinonitrile Analogues of Crolibulin and CA-4

Liu et al. recently published a series of novel 4,6-diphenyl-2-(1H-pyrrol-1-yl)nicotinonitrile analogues of crolibulin (**78**, Figure 11) and CA-4 (**2**, Figure 3) using a 2-(1*H*-pyrrol-1-yl)pyridine ring (highlighted in blue, **79**, Figure 12) as a link bridge in order to retain the *cis*-orientations of A and B rings. Despite crolibulin’s potent pro-apoptotic and VDA activity, its neurological and cardiovascular toxicity may limit its clinical use. Consequently, efforts have been made to design novel analogues based on the crolibulin scaffold. SAR studies indicated that crobulin’s cyano group and A-ring were of utmost importance for anti-tumour action. While these moieties were retained, crolibulin’s chromene structure was replaced with a phenyl substituted pyridine ring. Construction of these analogues was based on detailed SAR studies of the restricted CA-4 analogues. Considering the structural similarities required for effective CBSIs, the trimethoxyl fragment was reintroduced to ensure optimal binding to the CBS. An additional structural feature; a fluorine atom was introduced into the B-ring, further enhancing antiproliferative activity.

Several monofluorinated derivatives including **79** were over 300 times more potent than difluorinated compounds with IC_50_ values ranging from 30 nM to 310 nM across five human cancer cell lines, comparable to CA-4 and superior in potency than crolibulin. Compound **79** was more potent than CA-4 and crolibulin. The introduction of the 2-(1*H*-pyrrol-1-yl) pyridine in place of CA-4’s double bond not only maintained anti-tumour potency by preventing *cis-trans* isomerization but enhanced it. Molecular docking studies illustrated that **79** may bind in the CBS in a novel and superior mode, which authors note deserves further investigation [146]. Compound **79** may be advanced for further clinical evaluation based on preliminary comparative pre-clinical data between it and crolibulin [146].

### 5.11. Indenes CP248 and CP461

Two potent derivatives of the pro-apoptopic sulindac-derivative exisulind (**80**, Figure 12), CP248 (also known as OSIP 486823, **81**) and CP461 (also known as OSI-461, **82**, Figure 12) (Table 1), have been found to cause growth inhibition and apoptosis in human carcinoma cell lines. Together these compounds represent a new novel class of compounds termed selective apoptotic antineoplastic drugs (SAANDS). These compounds show promise due to selective induction of apoptosis in cancerous and pre-cancerous cells. Compound **81** is an MTA, disrupting microtubule polymerisation and perturbing spindle function causing cell death in G_2_/M phase. The trimethoxyphenyl group (highlighted in blue, Figure 12) of this molecule plays a role in reversible binding to the CBS, competing with colchicine likely because this portion of the molecule resembles the A ring of colchicine [147]. Compounds **81** and **82** are reported to be 100-1000-fold more potent than exisulind in terms of the inhibition of cell growth and induction of cancer cell apoptosis [148]. Compound **81** is extremely potent in colon cancer cells with an IC_50_ value of 3 nM. Similarly, **82**, despite the absence of the trimethoxyphenyl ring structure, has similar effects on M-phase cell cycle arrest, microtubule depolymerisation and the inhibition of mitotic spindle formation. Compound **82** appears to interact with tubulin in a different mechanism to **81** and the actual binding site remains to be determined. It has successfully undergone and is currently in various clinical trials including phase II trials for the treatment of renal cell carcinoma, prostate cancer and chronic lymphocytic leukaemia, sponsored by Astellas. The compound has yet to be marketed but it appears to have a promising future [149].

## 6. CBSIs in Clinical Trials and Clinical Use

### 6.1. Combretastatin A-4 Analogues in the Clinic

Translation of CA-4 to clinical use has been hindered by two main obstacles: isomerisation to the less potent *trans* isomer and poor water solubility. CA-4 has been an extremely attractive lead molecule for decades for the treatment of solid tumours and therefore many attempts to create diverse analogues of CA-4 are documented in the literature. The vascular damaging properties of CA-4 have prompted clinical evaluation of CA-4 phosphate derivatives (e.g., CA-4P, **83**, Figure 13). These prodrugs require chemical conversion by metabolic enzymatic processes into active pharmacological agents. Pettit et al. were the first to describe data for phosphate prodrugs of both CA-4 and CA-1 [150]. Initial attempts to synthesise prodrug derivatives of CA-4 involved the addition of a serinamide (highlighted in blue, Figure 13) to the B ring’s hydroxyl group, creating an analogue known as AVE8062 (**85**, Figure 13). This CA-4 amino acid prodrug was marketed by Sanofi as ombrabulin. The serine is cleaved by aminopeptidases in vivo to form the active derivative [151]. The most successful prodrugs of CA-4 to date are the phosphate salt derivatives CA-4P (fosbretabulin, **83**, Figure 13) and CA-1 diphosphate (Oxi-4503, **84**, Figure 13). These prodrugs were converted to the active counterparts seven times faster in tumour and liver preparations than in blood [152]. These prodrugs are the preferred molecules for further development and clinical trials.

#### 6.1.1. Fosbretabulin—A CA-4 Analogue Prodrug

Several phase I/II/III clinical trials have been completed using CA-4P (fosbretabulin, **83**, Figure 13) (Table 2) for anaplastic thyroid cancer, non-small-cell lung cancers, relapsed ovarian cancer and advanced solid tumours as tumour vascular disrupting/targeting agent. It is rapidly metabolised into combretastatin after administration by non-specific endogenous phosphatases. Compound **83** has been trialled as monotherapy in clinical trials and retained its VDA in cancer patients at doses that are well tolerated. Its short half-life and reversible effects mean that it does not exhibit the traditional side-effects of tubulin-targeting and anti-angiogenic agents such as proteinuria and haemorrhage and may prolong survival in cancer patients [153]. To date, **83** has not been approved for therapeutic use due to its failure as a monotherapy, conformational instability and short half-life [154]. Due to limited success as monotherapy, further clinical evaluation is underway using **83** in combinatorial regiments with approved cancer therapies [53,54]. It has gained orphan drug status granted by the EMA in August 2013 for the treatment of ovarian cancer. At the time of designation in the EU, Diamond Biopharm Limited provide information that fosbretabulin tromethamine improved response and outcomes for platinum-resistant ovarian cancers in combination with carboplatin and paclitaxel [155]. As of March 2016, it has gained orphan drug approval for the treatment of gastro-entero-pancreatic neuroendocrine tumours. (GEP-NETS). While several products are authorised for the symptomatic management of GEP-NETs, current data shows positive results in response to fosbretabulin tromethamine [156]. Given the potential clinical success of **83** it will be exciting to see if any of the structurally modified CA-4 analogues discussed in this review make it to the clinic in the coming years.

#### 6.1.2. Oxi4503—A CA-1 Prodrug

Oxi4503 (**84**, Figure 13) (Table 2), the diphosphate derivative of combretastatin A-1 (**3**, Figure 3), received fast-track orphan drug approval by the FDA in 2017 and is in development for the treatment of acute myeloid leukaemia (AML) by Mateon Therapeutics (formerly known as Oxigene). Despite inducing tumour necrosis, all VDAs will leave a layer of viable tumour tissue in tumour periphery, which survive due to nutritional support from normal blood vessels. This viable tumour tissue may induce rigorous and reactive angiogenesis which continues to support tumour re-growth. Combining an anti-angiogenic agent with VDA therapy may counteract this phenomenon.

Such combination therapy was trialled using the anti-angiogenic agent bevacizumab (Avastin) and **84**. A combination of bevacizumab and **84** was better than either treatment alone [40]. Treatment with **84** appears to sensitise leukaemic tumour vessels to bevacizumab. Currently, development is ongoing for the treatment of AML and myelodysplastic syndrome (MDS). It is now known that **84** targets leukaemia through a dual mechanism of action. It firstly disrupts the shape of tumour bone marrow endothelial cells in similar mechanism to regular VDAs, which in turn releases bone marrow endothelial cells. This makes them more freely available and vulnerable towards regular chemotherapeutic agents. Therefore, it holds greatest potential in combination regimens. Compound **84**, like 3, forms an orthoquinone cytotoxic mediator in vivo which acts as a myeloperoxidase activator, directly killing tumour cells. A Phase Ib/II clinical trial, initiated in 2015, is underway investigating **84** in combination with cytarabine for treatment of MDS and AML [157].

#### 6.1.3. Ombrabulin/AVE8062—An CA-4 Amino Analogue Prodrug

Another CA-4 analogue ombrabulin/AVE8062 (**85**, Figure 13) (Table 2) marketed by Sanofi Aventis with superior solubility and oral bioavailability in comparison to CA-4 has been trialled *Ombrabulin*/*AVE8062* in Phase 2 clinical studies for the treatment of ovarian carcinoma. It has reduced toxicity and improve anti-cancer activity relative to CA-4 [158]. It was designated orphan drug status in 2011 for the treatment of soft tissue sarcoma due to its benefit as an alternative treatment where existing protocols fail. Therapeutic synergy with cisplatin was evident in pre-clinical and phase I studies [159] and so its orphan drug approval encompasses inclusion in combination chemotherapy regimens to improve outcomes versus monotherapy alone. However, during a double-blind phase III trial for advanced soft tissue sarcoma, a progression free survival result of less than one week was considered statistically insignificant. Incremental efficacy resulted in Sanofi withdrawing orphan status and has resulted in cessation of clinical development [160].

### 6.2. Non-Combretastatin A-4 Analogues in Clinical Trials

#### 6.2.1. Tivantinib

Tivantinib (**86**, Figure 13) (Table 1) was first reported as a selective oral inhibitor of MET, the tyrosine kinase receptor [161] for binding of MET to factors which promote pathways involved in tumourigenesis and metastasis [162]. It has been shown that tivantinib binds competitively with higher affinity than colchicine at the CBS pocket. Aoyoma et al. demonstrated that **86** was potent against MDR human cancer cell lines which resulted in many ongoing and completed clinical trials [163] It has been trialled in phase III trials against hepatocellular carcinoma (HCC), non-small-cell lung carcinoma (NSCLC) and liver cancer [164]. In May 2018, the results of a phase III, randomised, double-blind, placebo-controlled trial were published. Authors noted the urgent requirement for second line therapies for the treatment of HCC with overall median survival of patients being nine months and only 10% survival after five years [165]. Unfortunately, tivantinib failed to improve overall survival compared with placebo in patients with MET-high advanced HCC previously treated with sorafenib, a licensed anti-angiogenic treatment for HCC [166,167]. However, a systematic review published in 2017, which included 1824 patients from six randomized controlled trials, demonstrated that tivantinib showed significant improvement in progression free survival in solid tumours, and overall survival in particular with those patients that had high MET expression levels. Additionally, it appeared well tolerated by patients. Unfortunately, after multiple failed studies Kyowa Hakko Kirin Co. Ltd. have discontinued its development [168].

#### 6.2.2. Plinabulin (BAL27862)

Plinabulin (BAL27862, **75**, Figure 11) (Table 1) is a small-molecule MTA and is particularly active against Kirsten rat sarcoma viral analogue homologue (KRAS)-driven cancers, which are present in 30% of cancers and are associated with poor patient survival. The effectiveness of chemotherapy is limited in KRAS-driven cancers and is often associated with toxicity [169]. Pinabulin is a non-conventional CBSI derived from marine sources. It is an analogue of halimide, an aspergillus-derived natural product MTA. Plinabulin interacts with β-tubulin as a tubulin polymerization inhibitor/de-stabilizer. Pinabulin is currently in Phase III clinical trials for the treatment of stage IIIb/IV NSCLC, pioneered by Beyond Spring Inc. in combination with docetaxel. It is being trialled in epidermal growth factor receptor (EGFR) wild-type patients, which is an under-served patient population who typically fail to respond to conventional platinum-based regimens. EGFR patients typically have shorter mean survival times of 8–10 months versus 18 months for EGFR mutant NSCLCs. Studies are ongoing and if combination therapy proves more efficacious with an improved safety profile and extended quality of life for this cohort, it holds promise to become the preferred alternative treatment where standard first line regimens fail for the treatment of NSCLC [170].

#### 6.2.3. Lisavanbulin (BAL101553)

The amino acid lysine water-soluble prodrug of BAL27862/plinabulin (**75**, Figure 11), named lisavanbulin (also named BAL101553, **87**, Figure 13) (Table 1), is now in various clinical trials. It has completed its phase I study in adults and is now in Phase I/IIa [171].

Phase I trial results of lisavanbulin in patients with advanced solid tumours that are resistant to first line treatment indicate a loss of capillaries, suggesting that the compound has a vascular disrupting effect. Two patients of 16 demonstrated stable disease (laryngeal and rectal cancer) following > 16 cycles over 16 weeks. Results indicate that **87** is readily converted to **75** and is well tolerated at doses up to 60mg/m^2^ while having evidence of anti-tumour activity [172]. Lisavanbulin **(87)** has undergone initial testing in a paediatric preclinical testing program (PPTP). Vincristine is the only MTA commonly used in children. Since acquired drug resistance to vincristine is a large problem in this population, is has been recognised that alternative MTAs are needed. Given the capacity to circumvent drug resistance using 87, it was evaluated across 23 cell lines in the PPTP in vitro panel and also in vivo using PPTP solid tumour xenograft models. Unfortunately, efficacy was modest using 87 despite the parent compound having an in vitro IC_50_ of 13.8 nM. In xenograft models only 11% of cell lines exceeded the control. The reasons for this lack of efficacy remain unclear [173]. Currently, 87 is in phase I trials in combination with radiotherapy for treating patients with newly diagnosed glioblastoma with the primary objective of determining overall progression free survival [174]. However, this amino acid prodrug is a promising emerging anti-cancer agent in clinical development, which may translate to clinical use in the coming years.

#### 6.2.4. Crolibulin

Crolibulin (**78**, Figure 11) (Table 1) has been in Phase I/II Clinical trials since November 2010 for anaplastic thyroid cancer in combination with cisplatin. Unfortunately, the phase II trial has not yet been completed due to recruitment issues; although the compound shows potential [175].

## 7. Conclusions

MTAs have achieved great success in treating many diverse forms of cancer. However, acquired drug resistance over the course of treatment has become a significant limiting clinical factor with typically used agents such as the taxanes and vinca alkaloids susceptible to P-gp-mediated drug efflux. While colchicine itself cannot be used in the treatment of cancer due to systemic toxicity, several other CBSIs show promise in the future treatment of cancers of all subtypes, including those with acquired MDR characteristics. Many of these CBSIs have potent anti-angiogenic and anti-vascular activity. Hundreds of analogues with promising in vivo data targeted towards MDR cancers are known. Further progression towards clinical studies is a limiting step for many reasons, including the funding required to do so. Extensive pre-clinical studies, as outlined throughout this review, strongly suggest that CBSIs show potential to augment the use of typically used agents, thereby surmounting drug resistance mediated by P-gp, MRP1 and MRP2. Efflux pumps. There are currently a number of CBSIs in clinical trials and the results are likely to dictate the future for this drug class. It is likely that we will see new CBSIs come to the market for clinical use in the future. It is time to take CBSIs from the chemistry bench to the clinic.

## Figures and Tables

**Figure 1 pharmaceuticals-13-00008-f001:**
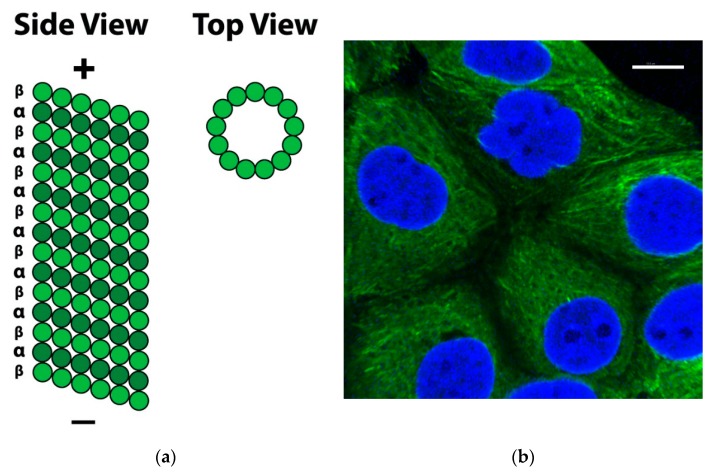
Microtubule structure from side and top perspectives. (**a**): Side View: Long, linear protofilaments consisting of αβ-tubulin heterodimers associate laterally; Top View: Association of 13 protofilaments forms the microtubule, a long hollow cylinder. (**b**): Microtubule network in MCF-7 breast cancer cells. Cells were stained with mouse monoclonal anti-α-tubulin−fluorescein isothiocyanate (FITC) antibody (clone DM1A) (tubulin, green) and Alexa Fluor 488 dye, then counterstained with 4′,6-diamidino-2-phenylindole (DAPI) (nuclei, blue). Scale bar: 10 μM.

**Figure 2 pharmaceuticals-13-00008-f002:**
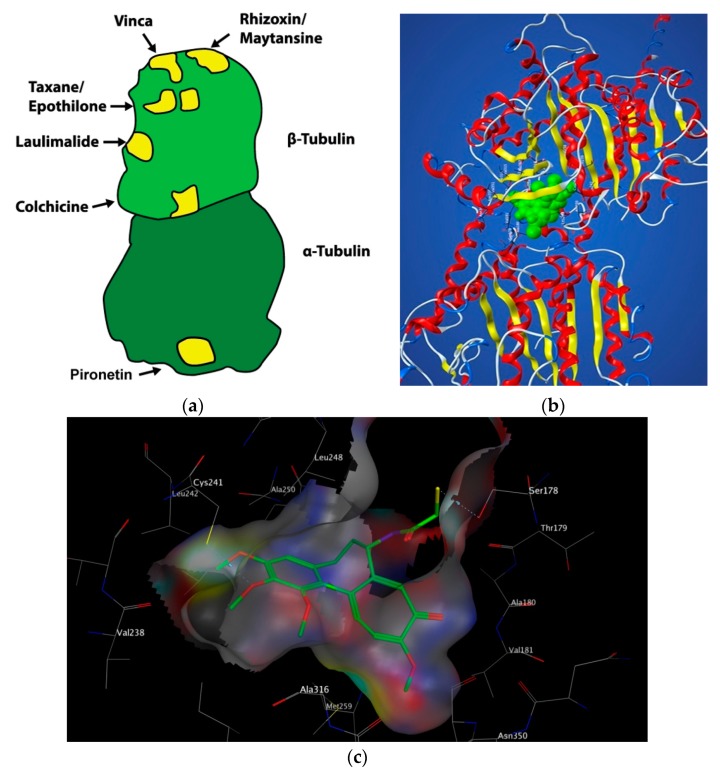
(**a**): Schematic representation of the different drug-binding sites on tubulin. (**b**): Space-filling model of colchicine (green) in the colchicine-binding site (CBS). (**c**): Model of N-deacetyl-N-(2-mercaptoacetyl) colchicine (DAMA)-colchicine bound at the interface of αβ-tubulin. Key binding interactions are with Thr179 and Val181 of α-tubulin and Cys241 and Asn258 of β-tubulin. Colour key: green = carbon; red = oxygen; blue = nitrogen; yellow = sulphur.

**Figure 3 pharmaceuticals-13-00008-f003:**
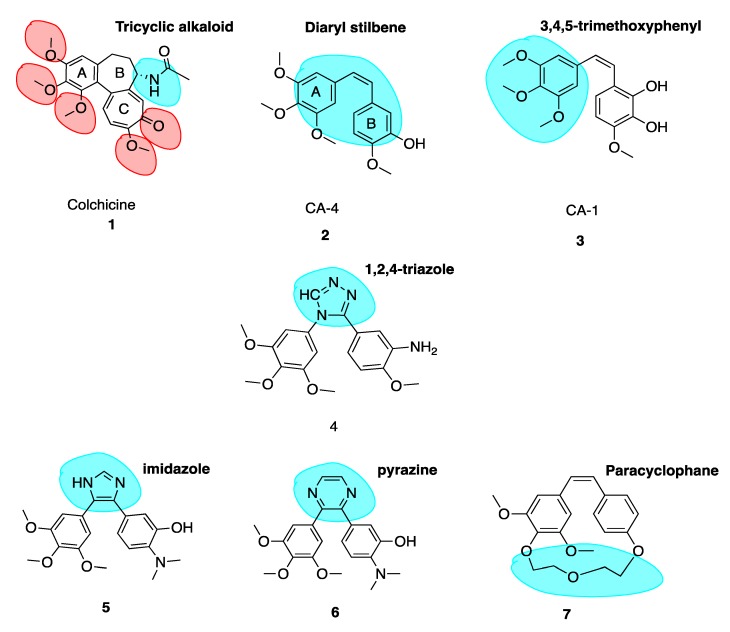
Structures of colchicine, combretastatin A-4, combretastatin A-1 and analogues 4–7.

**Figure 4 pharmaceuticals-13-00008-f004:**
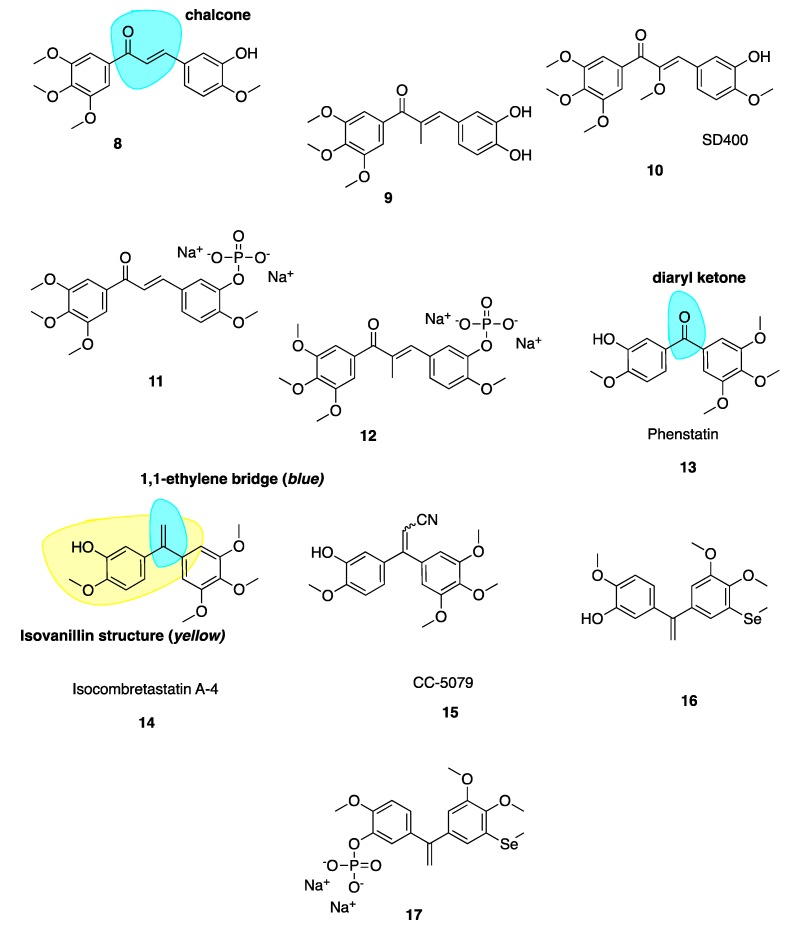
Combretastatin A-4 analogues **8**–**17**.

**Figure 5 pharmaceuticals-13-00008-f005:**
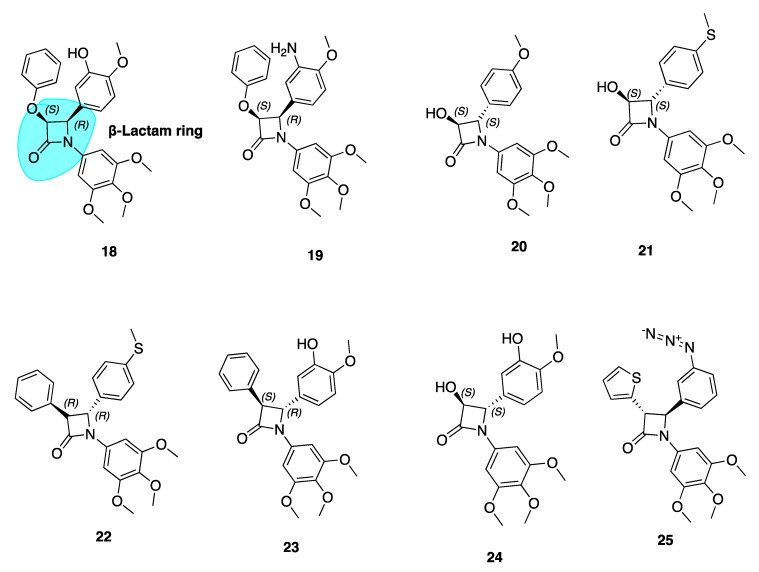
Structures of β-Lactams **18**–**25**. (*Cis* isomers of **18** and **19** and *trans* isomers of **20**–**24** isolated only. (One enantiomer represented for each compound).

**Figure 6 pharmaceuticals-13-00008-f006:**
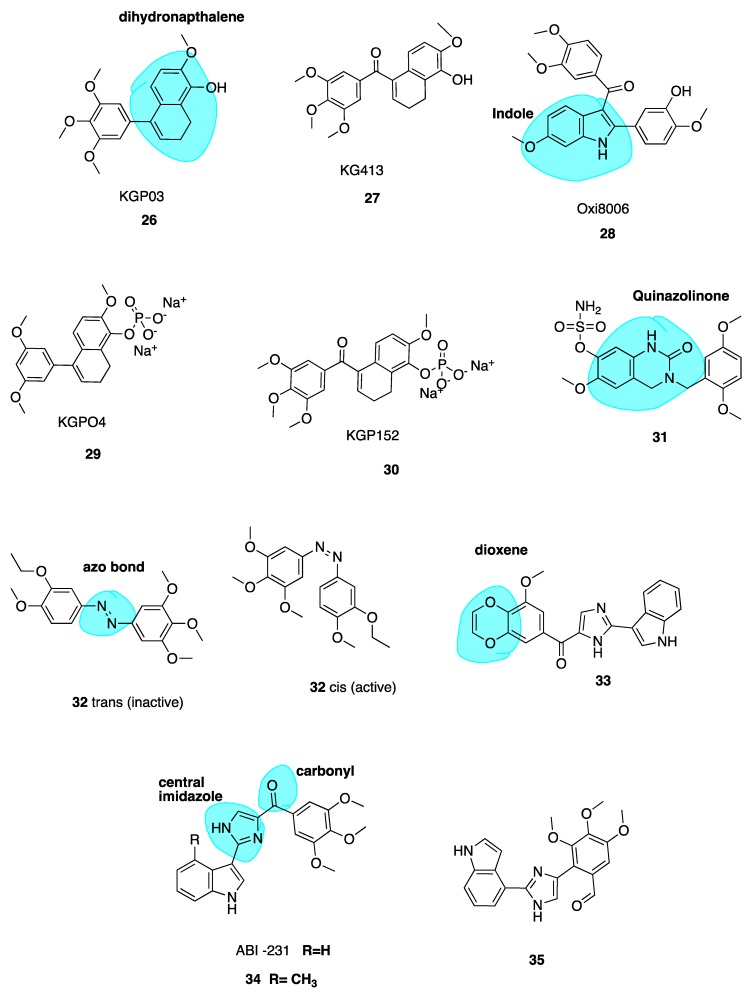
Structures of compounds **26**–**35**.

**Figure 7 pharmaceuticals-13-00008-f007:**
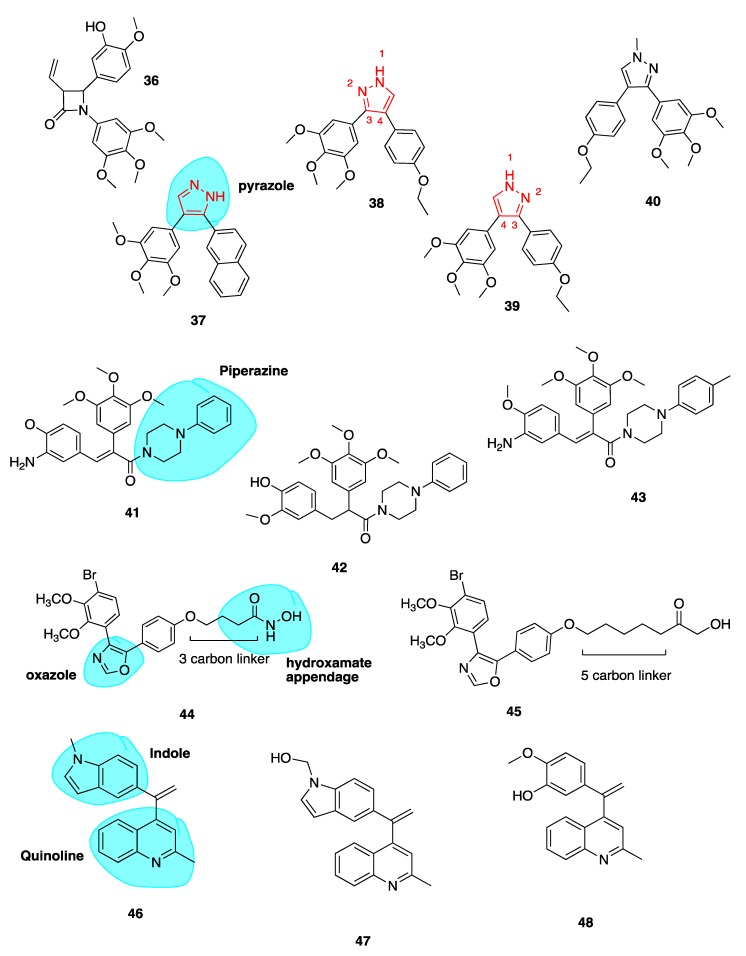
Structures of combretastatin analogues **36**–**48**.

**Figure 8 pharmaceuticals-13-00008-f008:**
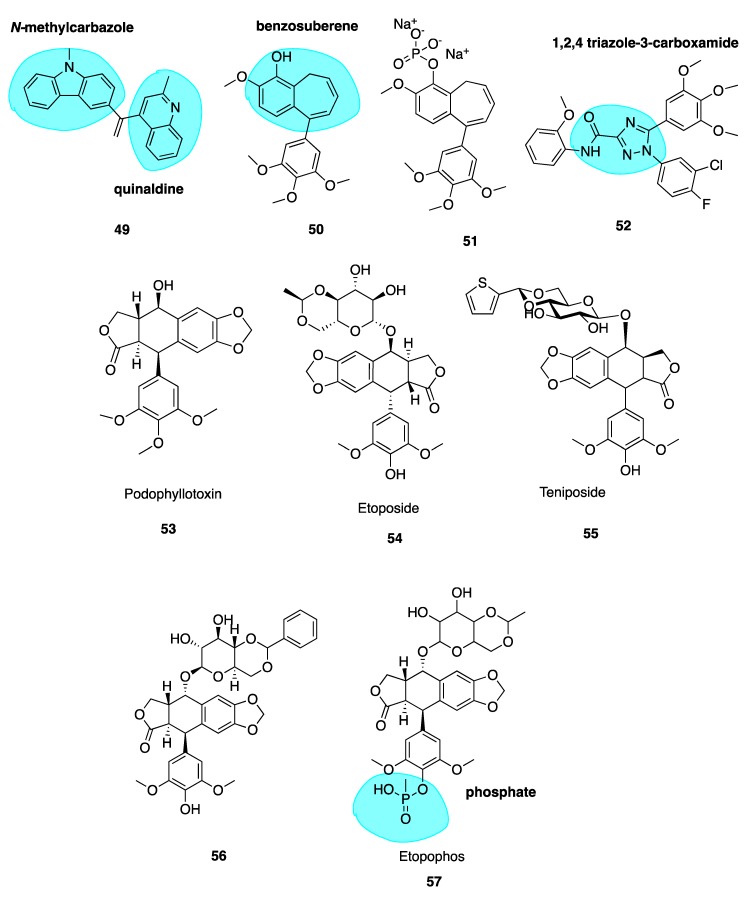
Structures of heterocyclic *iso*CA-4 derivative **49**, novel benzosuberene analogues **50** and **51**, carboxamide derivative **52**, podophyllotoxin **53**, etoposide **54** and analogues **55**–**57**.

**Figure 9 pharmaceuticals-13-00008-f009:**
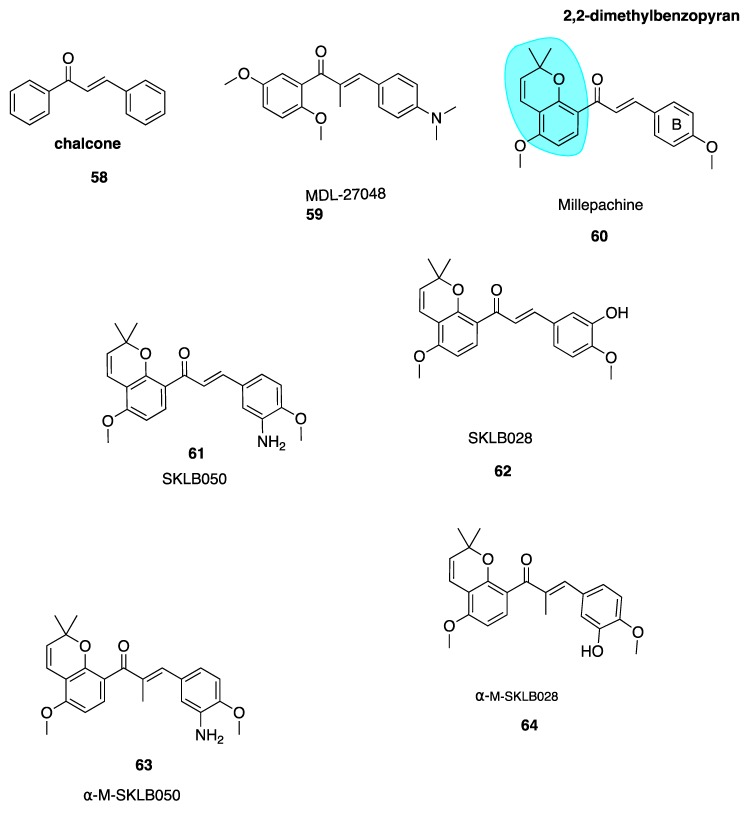
Structures of chalcones **58**–**64**.

**Figure 10 pharmaceuticals-13-00008-f010:**
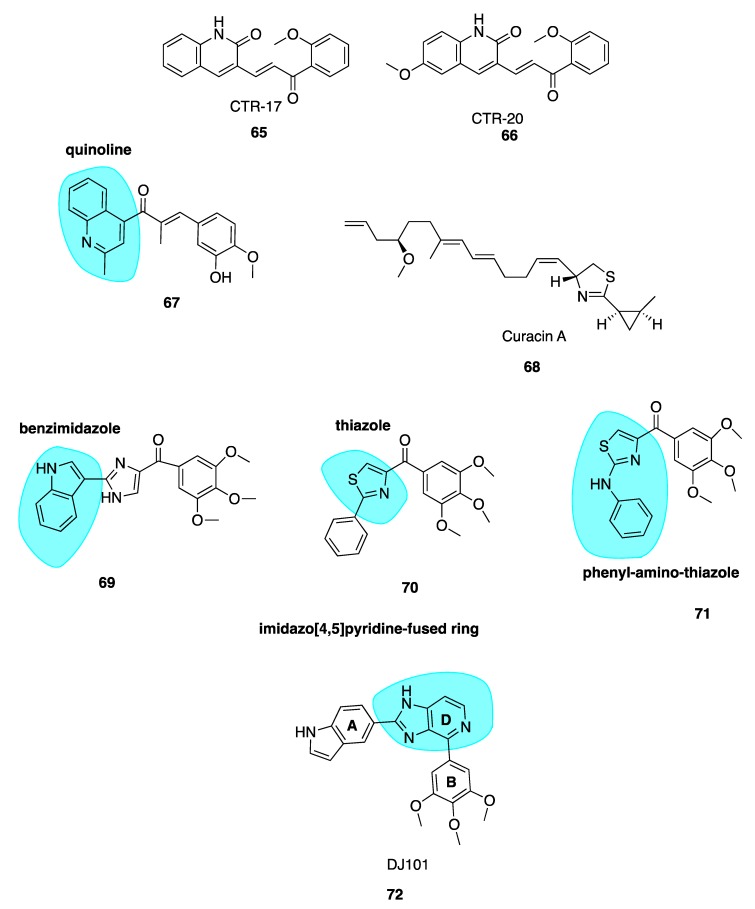
Structures of colchicine-binding site inhibitors **65**–**72**.

**Figure 11 pharmaceuticals-13-00008-f011:**
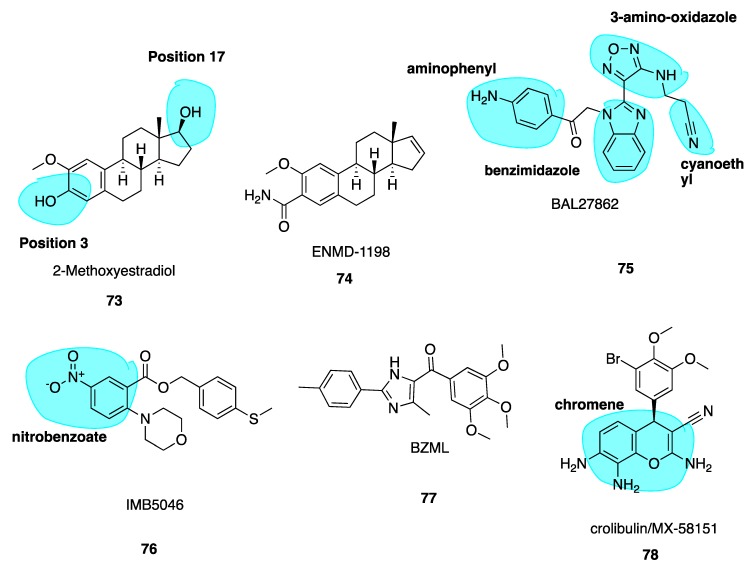
Structures of compounds **73**–**78**.

**Figure 12 pharmaceuticals-13-00008-f012:**
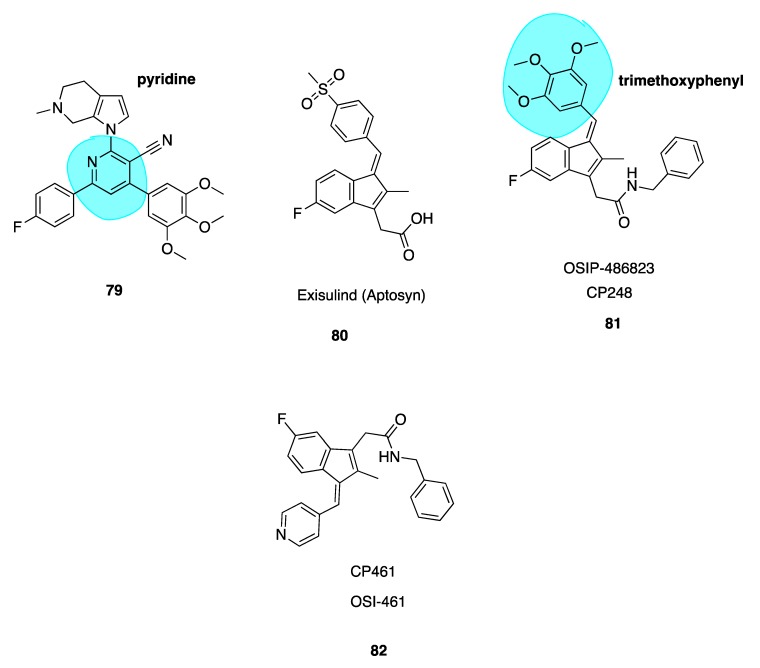
Crolibulin derivative **79**, exisulind, and exisulind derivatives **81**–**82**.

**Figure 13 pharmaceuticals-13-00008-f013:**
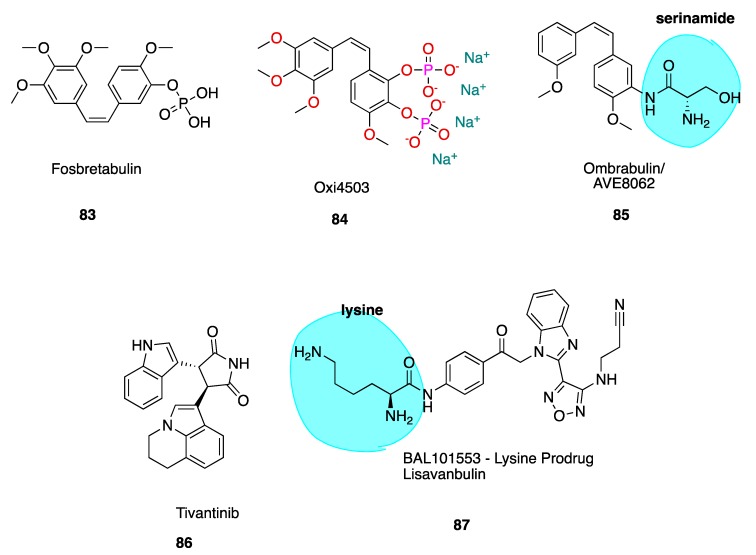
Structures of colchicine-binding site inhibitors in clinical trials.

**Table 1 pharmaceuticals-13-00008-t001:** CBSIs in clinical trials and clinical use—non combretastatin A-4 analogues.

Drug Name	Section	Structural Features	Stage in Clinical Trials and Disease Treated	Company Developing Drug
ENMD1198 (**74**)	Section 5.5	2-Methoxyestradiol derivative	Phase I refractory solid tumours	Casi pharmaceuticals (no data published in recent years)
CP461/OSI-461 (**82**)	Section 5.11	Derivative of exisulind	Phase II Renal cell carcinoma Prostate cancer Chronic lymphocytic leukaemi (Not yet marketed or in clinical use)	Astellas
Tivantinib (**86**)	Section 6.2.1	Heterocyclic fused ring system	Phase II Hepatocellular carcinoma (HCC) Liver cancer Non-small-cell lung carcinoma (NSCLC)	Kyowa Hakko Kirin Co. Ltd. (development discontinued in 2018 following multiple poor clinical trial outcomes)
Plinabulin (BAL27862) (**75**)	Section 6.2.2	Contains benzimidazole and 3-amino oxadiazole moieties	Phase III stage IIIb/IV NSCLC (in combination with docetaxel)	Beyond Spring Inc.
Lisavanbulin (BAL101553) (**87**)	Section 6.2.3	Lysine prodrug of Plinabulin	Phase I/IIa advanced solid tumours, refractory to standard therapy	Basilea Pharmaceutica
Crolibulin (**78**)	Section 6.2.4	Chromene derivative	Phase I/II Clinical trials Anaplastic Thyroid Cancer	Immune Pharmaceuticals Inc; National Cancer Institute (USA) (clinical trial progression limited due to recruitment issues)

**Table 2 pharmaceuticals-13-00008-t002:** Summary of colchicine-binding site inhibitors (CBSIs) in clinical trials and clinical use—combretastatin A-4 analogues.

Drug Name	Section	Structural Features	Stage in Clinical Trials and Disease Treated	Company Developing Drug
Fosbretabulin (**83**)	Section 6.1.1	Phosphate prodrug of CA-4	Phase I/II/III Anaplastic thyroid NSCLC Relapsed ovarian cancer (orphan drug approval granted August 2013) Gastro-entero-pancreatic neuroendocrine tumours (GEP-NETs) (orphan drug approval granted March 2016)	Diamond Biopharm Limited
Oxi4503 (**84**)	Section 6.1.2	DiPhosphate prodrug of CA-1	Phase Ib/II acute myeloid leukaemia	Mateon Therapeutics (previously Oxigene)
Ombrabulin/AVE8062 (**85**)	Section 6.1.3	Serine prodrug of CA-4	Phase II Ovarian cancer Advanced soft tissue sarcoma (unsuccessful—orphan drug status withdrawn)	Sanofi Aventis

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
