# Peer review of "Colchicine-Binding Site Inhibitors from Chemistry to Clinic: A Review"

_pharmaceuticals, 2020, doi:10.3390/ph13010008_

Round 1

Reviewer 1 Report

Overview:

This manuscript reviews the latest developments on colchicine binding site analogues. It is very detailed and comprehensive, providing a history of colchicine site derivatives produced and introducing the newest ones from 2017 – 2019. Most are combretastatin analogues. It finishes with a summary of the colchicine binding site inhibitors that are in clinical trials. Three are FDA-approved and in the clinic for anticancer treatment (etoposide, teniposide, and etopophos) but do have some side effects. Seven others are in various stages of clinical trials. The manuscript is well-written and an excellent reference for the colchicine binding site compounds, providing the rationale for design of analogues, and strengths and shortcomings of new analogues. Preclinical and clinical testing results are provided.

Major Points:

The paper is an excellent, comprehensive review, discussing the growth inhibition potencies, vascular disrupting effects, and any side effects of the analogues. It needs some minor typographical and grammatical corrections that are detailed below. Perhaps for the non-chemist, it would help if the different chemical moieties added to a lead structure were identified by an insert in the figure or as a separate figure showing the major groups, for example, chalcone, lactam, piperazine, indole, quinoline, heterocycles, etc. Figure 2. The diagram of the binding sites on the heterodimer could include the α-tubulin binding site for pironetin. In addition, including a colchicine chemical structure and the amino acids in α- (Thr179, Val181) and β-tubulin (Cys-241, Asn258) that bind colchicine would be useful for following the design rationale that is discussed in the review. On page 31, Line 1033, the text states that there are currently 3 CA-4 derivatives and two non-CA-4-like CBSIs in clinical trials. From the data in the paper, there appears to be four CA-4 derivatives and three non-CA-4 derivatives in clinical trials. The CA-4 ones are crolibulin, fosbretabulin, Oxi4503 dephosphate, and lisavanbulin, and the non-CA-4 ones are ENMD-1198, plinabulin, and OSI-461.

Minor Points

Figure 1. Since the plus end cap ends in beta-tubulin and the minus end in α-tubulin, it would make more sense to have β-tubulin at the top of the Side View as well as in the Top View. Page 16, Line568 “isovanillin moieties” are mentioned; however, they have not been referred to earlier in the paper. The numbering of sections could be improved. Most are in section 4. Section 4 could be broken up to: Previous Colchicine-Binding Site Inhibitors New Colchicine Binding Site Inhibitors in Last Three Years CBSIs in Clinical Trials Conclusion

Typographical or Style Errors 

Page 1 Abstr

              Line 8   discovery of microtubules, and they have

Page 2   L62    form a 24 nm diameter                     

                 L68    cancer cells in interphase; cells

                 L70    green) Alexa

Page 3   L79    hydrolysis to GDP weakens

              L81    pole is retained and microtubules

              L94    diverse MTAs which have

              L96    vast majority act by

              L97    during G2M of the cell cycle.

              L100  agents, including the taxanes, epothilones, and laulimalide

              L101  colchicine, the vinca alkaloids, and maytansine. This

Page 4   L122  It binds to tubulin in non-cancerous and causes impaired

              L129  αβ-subunits are lost and as

              L135  which has a width

              L137  β-tubulin, subsequently

              L142  Val-181 within α-tubulin

Page 6   L211  {32,33]. CA-4

Page 7   L234  although the precise

              L242  provided by the actin

              L273  Oshumi et al. were one of the first

Page 8   L280  combretastatin heterocycle derivatives

              L292  in an attempt to

              L318  IC50 of 4.3 nM

Page 11 L403  extremely high nanomolar potency

Page 13 L445  photo-responsive

              L452  One compound 32 (Figure 6)  32 should be bold

              L461  activity (33,34,35) (Figure 6).

              L?      Compound 33 is never directly referred to in the text

Page 15 L514  a pyrazole can substitute

Page 16 L551  HDAC inhibitors have shown shortcomings

Page 19 L642  phenyl rings joined by an

Page 21 L677  B ring caused a significant

              L694  which covalently bind to their

              L717  67 is an inhibitor of the MRP1

Page 22 L737  CA-4 with concentrations of 16 ug/mL attainable compared to

              L739  insert a space between “1” and “mg/mL”

              L747  Collectively these data would suggest 66 has potential for

Page 24 L797  antiangiogenic properties [138]. 72 is metabolised 

              L801  Of a series of analogues

Page 26 L843  MX-58151 (77, Figure 11) belonging    bold the “77, Figure 11”

              Also, many bold fonts for compounds are needed on pages 26 – 30

Page 28 L903  AVE8062 (84, Figure 13).

Page 29 L929  current data show positive

              L931  discussed in this review

              L960  double blind phase 3

              L962  incremental efficacy                                                                                                 

Page 30 L994  18 months for EGFR  

         L1012    exceeded the control. The reasons for this lack of efficacy remain

Pages 31 – 40 (References)                                                                                                                                                                                 Ref 4    GM, C.,

Ref 9     McCred1E

Ref 12   S. Y. C. M. Y. T. Y. O. V. H. O. U. M. S. H.,

Ref 31   SA, H.; GM, T.; GR, P.; DJ, C.,

Ref 45   Roberto Gaspari, A. E. P., Katja Bargsten,2 Andrea Cavalli, and Michel O. etc.

Ref 91   Journal of experimental & clinical cancer research

Ref 134    Peter, W.; Jonathan, T. R.; Billy, W. D.,

Ref 137    Kinsie E. Arnst, Y. W., Zi-Ning Lei, Dong-Jin Hwang, Gyanendra Kumar, Dejian etc

Ref 139    <em>in vivo</em>

Ref 151   GILLIAN M TOZER, C. K., CHARLES S PARKINS, and SALLY A HILL, Tozer, G.                       M.; Kanthou, 1484 C.; Parkins, C. S.; Hill, S. A.,

Ref 160      Jean-Yves Blay, Z. P., Anthony W Tolcher, Antoine Italiano, Didier Cupissol etc

Also, about 26% of references have the title in upper case first letters: Example Ref 13: “The Role of Colchicine in Acute Coronary Syndromes”

Author Response

We thank reviewer 1 for your review and kind comments about our manuscript on colchicine binding site inhibitors. We have made the suggested changes as detailed below.

Reviewer comment: The paper is an excellent, comprehensive review, discussing the growth inhibition potencies, vascular disrupting effects, and any side effects of the analogues. It needs some minor typographical and grammatical corrections that are detailed below. Perhaps for the non-chemist, it would help if the different chemical moieties added to a lead structure were identified by an insert in the figure or as a separate figure showing the major groups, for example, chalcone, lactam, piperazine, indole, quinoline, heterocycles, etc. Figure 2.

Author response: For the non-chemist, the functional groups mentioned in the text are now highlighted in blue with annotations made to the relevant figure in brackets in the text. 

Reviewer comment: The diagram of the binding sites on the heterodimer could include the α-tubulin binding site for pironetin.

Author response: We have modified figure 2 to indicate the α-tubulin binding site for pironetin.

Reviewer comment: In addition, including a colchicine chemical structure and the amino acids in α- (Thr179, Val181) and β-tubulin (Cys-241, Asn258) that bind colchicine would be useful for following the design rationale that is discussed in the review.

Author response: We have included a new image in figure 2 depicting the structure of DAMA-colchicine (from PDB entry 1SAO) docked in the colchicine-binding site of tubulin, with relevant amino acids shown.

Reviewer comment: On page 31, Line 1033, the text states that there are currently 3 CA-4 derivatives and two non-CA-4-like CBSIs in clinical trials. From the data in the paper, there appears to be four CA-4 derivatives and three non-CA-4 derivatives in clinical trials. The CA-4 ones are crolibulin, fosbretabulin, Oxi4503 dephosphate, and lisavanbulin, and the non-CA-4 ones are ENMD-1198, plinabulin, and OSI-461.

Author response: We have included two tables in the revised manuscript; one for CA-4 derived analogues in clinical trials and one for non-CA-4 analogues in clinical trials. We hope that this clarifies our subdivision of compounds.

Reviewer comment: Figure 1. Since the plus end cap ends in beta-tubulin and the minus end in α-tubulin, it would make more sense to have β-tubulin at the top of the Side View as well as in the Top View.

Author response: We have modified Figure 1 so that β-tubulin is at top of both views.

Reviewer comment: Page 16, Line568 “isovanillin moieties” are mentioned; however, they have not been referred to earlier in the paper.

Author response: With regards to isovanillin moiety, it is now outlined in blue in the parent IsoCA-4 compound and referred to in the text. 

Reviewer comment: The numbering of sections could be improved. Most are in section 4. Section 4 could be broken up to: Previous Colchicine-Binding Site Inhibitors New Colchicine Binding Site Inhibitors in Last Three Years CBSIs in Clinical Trials Conclusion

Author response: Sections have been modified. Section 4 is now broken into CA-4 analogues prior to 2017 followed by those published from 2017, 2018 and 2019. We have moved other analogues to separate sections.

Reviewer comment: Typographical or Style Errors

Author response: We have corrected these errors in the revised manuscript.

Reviewer 2 Report

Revised manuscript (pharmaceuticals-668605) presented interesting and valuable work regarding microtubule-targeting agents as potential or used in treatment drugs in many different forms of cancer. The authors have focused on compounds targeting the colchicine-binding site of tubulin, called colchicine-binding site inhibitors. There is a continued need for colchicine-binding site drug development because of exclusion of colchicine from treatment.

The review entitled “Colchicine-Binding Site Inhibitors from Chemistry to 2 Clinic: A Review” is well built. In introduction a reader can find out about importance of microtubules in mitosis in physiological conditions as well as in cancer. Mechanisms of polymerization and depolymerisation of microtubules were also described. Next two sections presented colchicine with its mechanism of action on tubulin, structure-activity relationships of colchicine derivatives binding to tubulin, and colchicine-binding site as a target for anticancer therapy. Section 4 is a wide description of the large number of colchicine-binding site inhibitors.

The authors are familiar with the topic of tubulin, they have published a number of papers on antitubulin inhibitors. The authors have knowledge as expert in this field.

The language of manuscript is adequate and the nomenclature is correct. Reviewer recommends manuscript to further publishing process after minor revision that is needed. 

Main points:

Section 4.6.1 is dedicated to beta-lactam-containing compounds published in one article. Addition of other examples could improve the review in this area. For example article: Dong-Jun Fu, Sci Rep. 2017; 7: 12788. “Structure-Activity Relationship Studies of β-Lactam-azide Analogues as Orally Active Antitumor Agents Targeting the Tubulin Colchicine Site” doi: 10.1038/s41598-017-12912-4 could be added. The authors should show in Fig. 8 structure of compound (1-(3-Chloro-4-fluorophenyl)-N-(2-methoxyphenyl)-5-(3,4,5-601 trimethoxyphenyl)-1H-1,2,4-triazole-3-carboxamide) described in 4.8.8. In reviewer opinion, it would be valuable to add Table including compounds which are clinically investigated. The Table could be a summary of section 4.20

Author Response

We thank reviewer 2 for your review and kind comments about our manuscript on colchicine binding site inhibitors. We have made the suggested changes as detailed below.

Reviewer comment: Section 4.6.1 is dedicated to beta-lactam-containing compounds published in one article. Addition of other examples could improve the review in this area. For example article: Dong-Jun Fu, Sci Rep. 2017; 7: 12788. “Structure-Activity Relationship Studies of β-Lactam-azide Analogues as Orally Active Antitumor Agents Targeting the Tubulin Colchicine Site” doi: 10.1038/s41598-017-12912-4 could be added.

Author response: We have added the Scientific Reports article by Fu et al. to section 4.3.1.2 and have added their most potent compound to Figure 5.

Reviewer comment: The authors should show in Fig. 8 structure of compound (1-(3-Chloro-4-fluorophenyl)-N-(2-methoxyphenyl)-5-(3,4,5-601 trimethoxyphenyl)-1H-1,2,4-triazole-3-carboxamide) described in 4.8.8.

Author response: The structure of this compound is now shown in Figure 8.

Reviewer comment: In reviewer opinion, it would be valuable to add Table including compounds which are clinically investigated. The Table could be a summary of section 4.20

Author response: We have added two tables for compounds which have progressed to clinical trials.